# ZeroGR: A Generalizable and Scalable Framework for Zero-Shot Generative Retrieval

**Weiwei Sun**[1, *]   **Keyi Kong**[2, *]   **Xinyu Ma**[3]   **Shuaiqiang Wang**[3]
**Dawei Yin**[3]   **Maarten de Rijke**[4]   **Zhaochun Ren**[5, †]   **Yiming Yang**[1]
[1]Carnegie Mellon University   [2]Shandong University   [3]Baidu Inc
[4]University of Amsterdam   [5]Leiden University
{sunnweiwei,luxinyayaya012,xinyuma2016.com,shqiang.wang}@gmail.com
yindawei@acm.org,m.derijke@uva.nl,z.ren@liacs.leidenuniv.nl,
yiming@cs.cmu.edu

## Abstract

Generative retrieval (GR) reformulates information retrieval (IR) by framing it as the generation of document identifiers (docids), thereby enabling end-to-end optimization and seamless integration with generative language models (LMs). Despite notable progress under supervised training, GR still struggles to generalize to zero-shot IR scenarios, which are prevalent in real-world applications. To tackle this challenge, we propose ZeroGR, a zero-shot generative retrieval framework that uses natural language instructions to extend GR across a wide range of IR tasks. Specifically, ZeroGR is composed of three key components: (i) an LM-based docid generator that unifies heterogeneous documents (e.g., text, tables, code) into semantically meaningful docids; (ii) an instruction-tuned query generator that generates diverse types of queries from natural language task descriptions to enhance corpus indexing; and (iii) a reverse annealing decoding strategy to balance precision and recall during docid generation. Furthermore, we introduce OpenInstIR, the most diverse open-source instructed retrieval dataset. We investigate the impact of instruction fine-tuning scale and find that performance consistently improves as the number of IR tasks encountered during training increases. Extensive experiments on the BEIR and MAIR benchmarks demonstrate that ZeroGR achieves competitive performance across a wide range of retrieval tasks, establishing a new state-of-the-art among GR methods. Our code is available at https://github.com/sunnweiwei/ZeroGR.

## 1 Introduction

Dense retrieval (DR) is arguably the most effective and widely adopted information retrieval (IR) paradigm today (Izacard et al., 2022; Karpukhin et al., 2020; Muennighoff et al., 2022; Thakur et al., 2021). It encodes documents and queries as embedding vectors. Despite its success, DR's expressivity is fundamentally limited by the embedding dimensionality (Cao et al., 2020) and does not fully use the capabilities of generative language models (LMs) (Tay et al., 2022). As an alternative, generative retrieval (GR) (Metzler et al., 2021) introduces a paradigm shift that encodes corpus information into the model parameters, enabling document retrieval by generating (relevant) document identifiers (docids). GR has demonstrated competitive performance on various IR tasks when large-scale supervised data is available (Chen et al., 2022a; Sun et al., 2023a; Tay et al., 2022), spanning both traditional web search (Campos et al., 2016) and knowledge-intensive retrieval applications (Petroni et al., 2020).

Despite its promising performance on in-domain tasks, GR still exhibits limited generalization to out-of-distribution IR tasks. Existing GR models are typically trained on specific corpora and queries, and prior studies have shown that such training leads to poor performance on unseen tasks (Liu et al., 2025b; Zhang et al., 2025c). In contrast, real-world IR models are typically evaluated in a broader

---

[*]Equal contribution.
[†]Corresponding author.

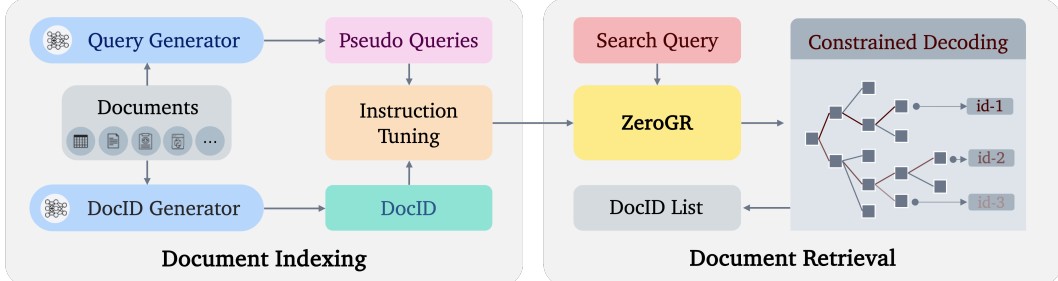

Figure 1: An overview of ZEROGR. Given a document collection, ZEROGR converts documents in the collection into unified DocID representations, generates diverse pseudo-queries, and builds a generative retrieval index. During online retrieval, ZEROGR decodes docids with reverse-annealed temperature scheduling to balance precision and recall.

setting, characterized by substantial diversity and heterogeneity. These often involve heterogeneous corpora and queries (Thakur et al., 2021), task-specific relevance criteria (Asai et al., 2022; Su et al., 2022), and predominantly zero-shot scenarios where no supervised data is available (Muennighoff et al., 2022; Thakur et al., 2021). Consequently, GR approaches designed for supervised conditions struggle to generalize to such heterogeneous and data-scarce retrieval scenarios.

To address the limitations of GR in zero-shot and heterogeneous IR scenarios, we propose **ZEROGR**, a generalizable framework for **ZERO**-shot **G**enerative information **R**etrieval. ZEROGR is a simple yet effective way to adapt GR to diverse IR tasks in a zero-shot setting by using natural language task instructions. Specifically, we advance GR along three dimensions: (i) for *docid design*, we propose a docid generator to efficiently convert a document of any format (e.g., paragraph, table, code) into a unified text-based docid representation; (ii) for *corpus indexing*, we propose an instructed query generator to generate diverse types of queries based on different task instructions; and (iii) for *docid decoding*, we propose a reverse annealing strategy that more effectively trades off precision and recall of docid decoding than prior work.

Building on ZEROGR, we investigate instruction fine-tuning along two key axes: the size of instruction tuning data and the size of the underlying model. Existing studies typically rely on single-domain (Weller et al., 2024) or closed-source training data (Muennighoff et al., 2024), hindering a systematic analysis of generalization. Instead, we collect the first large-scale open-source retrieval instruction dataset **OpenInstIR** (Open Instructed Information Retrieval Dataset), spanning diverse domains and tasks, enabling rigorous study of instruction generalizability in zero-shot retrieval. Through extensive experiment based on OpenInstIR, we find that increasing both the diversity and quantity of training tasks yields substantial improvements in zero-shot retrieval performance on unseen tasks. Beyond training data scaling, we also examine model size scaling and inference-time scaling for corpus indexing, observing consistently promising scaling trends in both cases. Our best-performing model, based on Llama-3B, outperforms previous generative retrieval methods and narrows the gap to state-of-the-art dense retrieval systems across heterogeneous IR benchmarks (Sun et al., 2024; Thakur et al., 2021). Notably, ZEROGR performs on par with OpenAI Embed-v3 on zero-shot MAIR tasks, highlighting its strong generalization to unseen retrieval tasks.

In summary, our contributions are as follows:

1. We propose ZEROGR, a zero-shot GR framework that can construct task-specific GR search indices based on natural language instructions.

2. Within ZEROGR, we enhance GR by introducing three key components: a unified text-based docid generator, an instruction-conditioned pseudo-query generator, and a reverse annealing decoding strategy.

3. We collect the first large-scale open-source instruction-tuning dataset for retrieval, OpenInstIR, that spans diverse domains and task formulations, enabling systematic study of model generalization in retrieval.

4. ZEROGR achieves competitive performance on heterogeneous IR benchmarks, establishing it as the first GR approach capable of generalizing to diverse tasks in a zero-shot setting.

## 2 RELATED WORK

**Document retrieval.** Document retrieval is a fundamental task in information retrieval, with broad applications in search engines and retrieval-augmented generation systems (Chen et al., 2025; Karpukhin et al., 2020; Lin et al., 2020). It typically follows a two-stage pipeline: an initial retrieval stage that recalls candidate documents, followed by a reranking stage for fine-grained ranking. Traditional sparse retrieval methods (Lafferty and Zhai, 2001; Robertson and Walker, 1997; Robertson and Zaragoza, 2009) rely on lexical overlap but suffer from vocabulary mismatch (Lin et al., 2020). Dense retrieval (DR) addresses this issue by embedding queries and documents into dense vectors and comparing them via inner product or cosine similarity (Karpukhin et al., 2020), with subsequent improvements from hard negative mining, late interaction, and pre-training (Izacard et al., 2022; Khattab and Zaharia, 2020; Qu et al., 2021; Wang et al., 2022a; Xiong et al., 2020). The reranking stage is usually performed using cross-encoders or LLM prompting (Chen et al., 2024; Liu et al., 2025a; Ma et al., 2023; Nogueira and Cho, 2019; Nogueira et al., 2020; Sun et al., 2023b; Zhang et al., 2025b).

However, this two-stage pipeline is difficult to optimize end-to-end due to its MIPS-based retrieval component and the objective mismatch with generative language model training (Bevilacqua et al., 2022; Tay et al., 2022).

**Generative retrieval.** Unlike traditional dense retrieval methods (Karpukhin et al., 2020; Xiong et al., 2020), GR formulates information retrieval as a docid generation task, enabling end-to-end optimization of the inference-time search index (Metzler et al., 2021; Tay et al., 2022). Previous research on GR has largely focused on three key aspects: (i) *Docid design*: Early approaches employed rule-based formats such as titles (Cao et al., 2020; Chen et al., 2022a), URLs (Zhou et al., 2022), or text spans/summaries (Bevilacqua et al., 2022; Li et al., 2023b). More recent work has shifted toward learning-based docid designs that capture corpus semantics more effectively, including embedding clustering (Tay et al., 2022) and RQ-VAE–based approaches (Wang et al., 2024; 2023b; Zeng et al., 2023). (ii) *Corpus indexing*: Several strategies have been explored to enrich corpus representations, such as document chunking (Tay et al., 2022), pseudo-query generation (Zhuang et al., 2022), rehearsal-based augmentation (Tang et al., 2023), multi-granular indexing (Wen et al., 2025), and continual training for dynamic corpora (Chen et al., 2023; Mehta et al., 2022; Zhang et al., 2025c). (iii) *Docid decoding*: The dominant approach has been constrained beam search (Cao et al., 2020; Tay et al., 2022). More advanced strategies include multi-stage decoding (Ren et al., 2023), multi-docid decoding (Li et al., 2023c), and simultaneous decoding (Zeng et al., 2024).

Despite steady progress, existing work primarily remains confined to supervised fine-tuning, relying heavily on training data and failing to generalize to zero-shot retrieval tasks.

**Instruction fine-tuning in IR.** Inspired by studies in LLM instruction tuning (Chung et al., 2022; Wang et al., 2022b), instruction fine-tuning for retrieval has gained increased attention to improve zero-shot IR performance (Asai et al., 2022; Su et al., 2022). Instruction-tuned models are able to adapt to various tasks based on natural language instructions that specify the relevance criteria. Recent studies in this direction include multi-task fine-tuning (Lee et al., 2024a), LLM-generated instruction data (Lee et al., 2024b; Oh et al., 2024; Wang et al., 2023a), and instruction-negatives (Weller et al., 2024). These efforts have primarily focused on dense retrieval or cross-encoder rerankers (Sun et al., 2024). To the best of our knowledge, we are the first to investigate instruction fine-tuning for GR and to conduct a systematic study of the factors that influence instruction fine-tuning.

## 3 PRELIMINARIES

**Zero-shot document retrieval.** We formulate the task of zero-shot document retrieval as follows. Given a corpus $\mathcal{D} = (d_1, \ldots, d_n)$ containing $n$ documents, a *corpus indexing* function $\mathcal{I}$ takes $\mathcal{D}$ as input and constructs a search index $m = \mathcal{I}(\mathcal{D})$. Then, a *retrieval* function $\mathcal{F}$ takes the index $m$ and a query $q$ as input, and returns a list of relevant documents: $(d_i, \ldots) = \mathcal{F}(m, q)$. Note that in a typical zero-shot document retrieval setting, no training data is available. However, a natural language task instruction $instr_t$ specifying the retrieval task is generally assumed to be available, as it is usually easier to obtain (Muennighoff et al., 2022).

In dense retrieval (Karpukhin et al., 2020), the indexing function can be defined as using a document encoder to encode the corpus as a embedding matrix such as $\mathbf{E} \in \mathbb{R}^{n \times k}$ and the index is defined as the matrix $m := \mathbf{E}$. Then for retrieval function, a query encoder encode the query $q$ as $\mathbf{q} \in \mathbb{R}^{1 \times k}$ and then perform maximum inner-product search (MIPS) over index $m$ to find closest document in embedding space.

**Generative retrieval.** GR aims to retrieve the document $d_i$ by generating the corresponding document identifier (docid) given the query $q$. To this end, GR assigns an identifier (docid) to each document in the corpus, e.g. $(z_1, \ldots, z_n)$, where each $z_i$ is a sequence of tokens $z_i = \{z_i^{(1)}, \ldots, z_i^{(T)}\}$ with a maximum length of $T$. Based on this, the indexing function $\mathcal{I}(\mathcal{D})$ of GR is to train a language model (LM) $\mathcal{M}$ on the corpus $\mathcal{D}$, encoding the corpus information and also document-docid mapping. The retrieval function $F$ is instantiated by the same $\mathcal{M}$, and it generates the relevant document identifiers (docids) $(z_1, \ldots, z_n)$ given the query $q$: $(z_i, \ldots) = \mathcal{M}(q)$.

## 4   ZEROGR

We propose ZEROGR, a zero-shot GR framework that can adapt LMs into task-specific generative search indexes based on task instructions. As shown in Figure 1, the ZEROGR framework consists of three key components: (i) a docid generator $G_\psi$, which takes a document $d_i$ as input and outputs its docid $z_i$; (ii) an instructed query generator, which takes a task instruction *instr* and a document $d_i$ as input and outputs multiple pseudo-queries; and (iii) a generative retriever $\mathcal{M}$, which takes the instruction and a query as input and generates a list of docids.

The ZEROGR pipeline proceeds as follows: (i) given a new corpus $\mathcal{D}$ and its associated task instruction *instr*, the docid generator assigns each document $d_i$ a docid $z_i$; (ii) the instructed query generator $G_\theta$ samples $B$ queries $\{q_{i,1}, \ldots, q_{i,B}\}$ for each document $d_i \in \mathcal{D}$, thereby creating $\langle q_{i,j}, z_i \rangle$ pairs; and (iii) the generative retriever is trained to predict the corresponding docid $z_i$ given the concatenation of *instr* and a sampled query $q_{i,j}$. After training, the generative retriever $\mathcal{M}(z \mid q, instr)$ serves as the search index $m$. For a given query $q$, a newly proposed reverse annealing decoding strategy is employed to generate a ranked list of docids as retrieval results.

### 4.1   UNIFIED DOCID REPRESENTATION

Documents in downstream IR tasks can be heterogeneous, e.g., financial tables (Zhu et al., 2022), code files (Liu et al., 2023), meeting transcripts (Golany et al., 2024), or legal cases (Bhattacharya et al., 2019). Existing simple docid strategies, such as using document titles, URLs, or spans (Bevilacqua et al., 2022; Cao et al., 2020), often fail to generalize to user-customized data. ZEROGR therefore introduces a model-based **docid generator** $G_\psi$ that maps any document to a short, keyword-rich sentence (typically 6–8 words) ranked by coverage. Formally, for a document $d_i$ we define the docid $z_i$ as follows:

$$z_i \;=\; G_\psi(d_i) \;=\; \arg\max_{t \in \mathcal{V}^{\leq L}} G_\psi\big(t \mid d_i\big), \tag{1}$$

where $t$ is a token sequence of length $\leq L$ (with $L = 8$) drawn from the vocabulary $\mathcal{V}$. To instantiate $G_\psi$, we first prompt a powerful LM (e.g., GPT-4o) to create a training set of $\langle d_i, z_i \rangle$ pairs. A smaller model (Llama-3.2-1B) is then fine-tuned on this data, enabling fast, scalable generation of unified docids across diverse IR tasks. Section 5.1 details our training data.

### 4.2   INSTRUCTED CORPUS INDEXING

Corpus indexing in GR encodes each document $d_i \in \mathcal{D}$ into the model's parameters so that, at inference time, the model can recover $d_i$ by *generating* its document identifier $z_i$. DSI-QG (Zhuang et al., 2022) accomplishes this by pairing every document with a set of pseudo-queries, but its effectiveness diminishes when the pseudo-query distribution diverges from real user queries (Dai et al., 2022; Pradeep et al., 2023). This gap is especially large in heterogeneous IR scenarios, such as conversational, code, or multimodal search.

We mitigate the distribution gap with an **instructed query generator** $G_\theta$, obtained by instruction-tuning a 1B-parameter Llama model on diverse IR datasets verbalized through task-specific instructions. Given a document $d_i$ and a task instruction *instr*, the generator produces a pseudo-query $q_{i,j}$

from the conditional distribution

$$q_{i,j} \sim G_\theta\big(\cdot \mid d, instr\big). \tag{2}$$

For each document we draw $B$ queries with a temperature of 1:

$$\mathcal{Q}_i = \{\, q_{i,1}, \ldots, q_{i,B} \,\}. \tag{3}$$

These $\langle d_i, z_i \rangle$ pairs are used to train the generative retriever $\mathcal{M}$ by minimizing the cross-entropy loss

$$\mathcal{L}(\phi) = -\sum_{d_i \in \mathcal{D}} \sum_{q_{i,j} \in \mathcal{Q}_i} \log \mathcal{M}\big(z_i \mid q_{i,j}, instr\big), \tag{4}$$

thereby embedding the corpus into the model's parameters. Table 4 summarizes the instruction-tuning datasets.

### 4.3 REVERSE-ANNEALED DOCID GENERATION

During inference, a GR model must decode each docid $z_i$ as a *sequence of tokens*. Standard beam search often collapses to a few high-probability sequences, hurting recall (Wu et al., 2025). We therefore propose **reverse-annealed sampling**: each $z_i$ is generated token-by-token, while the sampling temperature is gradually *increased* to encourage diversity. Let $f(\cdot)$ denote the trained decoder after corpus indexing, and let $T$ be a prefix tree whose leaves correspond to valid docids. For the $i$-th docid we decode a token sequence $\mathbf{x}_i = (x_{i,1}, \ldots, x_{i,L_i})$ using temperature $t_i = g(i)$. At position $j$ we sample $x_{i,j} \sim \mathrm{Softmax}\Big(\frac{\boldsymbol{\ell}_{i,j}}{t_i}\Big)\Big|_{T_{i,j}}$, where $\boldsymbol{\ell}_{i,j}$ are the logits conditioned on the current prefix $(x_{i,1:j-1})$, and the subscript $T_{i,j}$ masks probabilities to tokens that keep the prefix inside the tree. After the complete sequence $\mathbf{x}_i$ is produced, its leaf is removed from $T$ so no subsequent iteration can repeat the same docid. The per-iteration temperature $t_i$ follows a *normalized sigmoid*:

$$t_i = g(i) = T_{\max} \cdot \frac{\sigma\big(k(\frac{i}{K} - m)\big) - \sigma(-km)}{\sigma\big(k(1-m)\big) - \sigma(-km)}, \quad \sigma(z) = \frac{1}{1 + e^{-z}}, \tag{5}$$

where $K$ is the total number of docids to generate, $k > 0$ controls the slope, and $m \in (0, 1)$ sets the midpoint. Starting from a low temperature yields high-precision early selections; increasing $t_i$ over iterations boosts exploration, thereby balancing precision and recall across the final ranked list. See Algorithm 1 and Figure 6, both in the Appendix, for details.

## 5 EXPERIMENTAL DESIGN

Our experiments address the following research questions:

1. **How do model design and training strategies influence the performance of ZEROGR?** To answer this, we conduct a systematic study on the development set, investigating key factors in generative retrieval. Specifically, we analyze how instruction tuning task diversity (Section 6.1), docid design (Section 6.2), corpus indexing strategy, decoding strategy (Section 6.4), and model size (Section 6.5) affect performance.

2. **How does ZEROGR compare with dense retrieval methods?** We evaluate ZEROGR against leading models on the MAIR benchmark (Section 7.1) and conduct additional analysis on the BEIR datasets (Section 7.2).

### 5.1 OPENINSTIR: OPEN INSTRUCTED INFORMATION RETRIEVAL DATASET

Existing studies typically rely on single-domain (Weller et al., 2024) or closed-source training data (Muennighoff et al., 2024). To support the development of ZEROGR, we collect training data covering a diverse range of IR tasks. Specifically, we extract the training splits of public information retrieval datasets (e.g., those in (Muennighoff et al., 2024; Sun et al., 2024)) and construct a multi-task training set with annotated task instructions and relevance labels.

As shown in Table 1, **OpenInstIR** spans 69 IR tasks across 6 domains and contains 41 million query-document pairs. OpenInstIR is the largest open-source IR training corpus to date, offering greater domain and task diversity, detailed instructional annotations, and reliable relevance labels. See Appendix A for detailed statistics.

## 5.2 Evaluation datasets

To evaluate zero-shot GR on diverse down-stream tasks, we use the BEIR and MAIR benchmarks: (i) **BEIR** (Thakur et al., 2021). We evaluate models on all 12 tasks from the BEIR collection. (ii) **MAIR** (Sun et al., 2024). As we collect training data from a subset of MAIR tasks, we divide MAIR into seen and unseen subsets, where the unseen subset contains tasks not present in the OpenInstIR training data, to validate the zero-shot generalization of models. In constructing this benchmark, we curated a diverse set of long-tail tasks across 6 domains, and intentionally omitted redundant tasks (e.g.,

Table 1: Dataset domain statistics.

| Domain | OpenInstIR | | MAIR-Test | | BEIR | |
|---|---|---|---|---|---|---|
| | Tasks | Size | Tasks | Size | Tasks | Size |
| Academic | 18 | 744,160 | 5 | 500 | 2 | 200 |
| Code | 13 | 1,969,586 | 3 | 300 | 0 | 0 |
| Finance | 8 | 31,315 | 5 | 439 | 1 | 100 |
| Legal | 7 | 23,086,948 | 4 | 300 | 0 | 0 |
| Medical | 5 | 421,430 | 8 | 459 | 2 | 150 |
| Web | 18 | 15,319,445 | 13 | 935 | 6 | 569 |
| All | 69 | 41,572,884 | 38 | 2,933 | 11 | 1,019 |

different years of the same competition) and structurally complex ones (e.g., IFEval) that would introduce evaluation overhead. Given the large size of the MAIR dataset, we also develop a Dev subset of MAIR for model ablation (see the detailed task breakdown in Table 6). Note that our current evaluation focuses on tasks with moderately sized corpora.

## 5.3 Evaluation metric

We evaluate models using the following metrics: (i) *Top-1 accuracy*, which measures retrieval precision by checking whether the top-ranked document is relevant to the query; (ii) *nDCG@10*, evaluates the quality of the top-10 ranked results by considering both the relevance and position of retrieved documents; and (iii) *Recall@100*, which assesses recall by calculating the percentage of relevant documents retrieved within the top-100 ranked list.

## 5.4 Implementation details

We implement the three components of ZEROGR, i.e., query generator, docid generator, and final generative retriever, all with Llama-based LMs. For the docid generator, a Llama-1B-Instruct model is trained on our curated document-docid pairs for 5 epochs with a constant learning rate of 5e-5. Similarly, for the query generator, a Llama-1B-Instruct model is trained on the OpenInstIR training set for 5 epochs with a constant learning rate of 5e-5. For the generative retriever, the model is trained for each evaluated task on data generated by the query generator and docid generator, based on our "Document Indexing" workflow described in Figure 1.

## 5.5 Baselines

We evaluate ZEROGR against several representative IR baselines, spanning different retrieval paradigms to provide a comprehensive comparison.

1. For sparse retrieval, we adopt the classical term-based model **BM25**, implemented using the BM25S package (Lù, 2024).

2. For traditional dense retrieval models trained on a single task, we include **Contriever-MARCO**, **GTR-base**, and **GTR-Large**, all of which are pretrained or fine-tuned on the MS MARCO dataset (Izacard et al., 2022; Ni et al., 2021), representing a common practice in dense retrieval pipelines.

3. For multi-task-trained dense retrievers, we incorporate **E5-Base** and **E5-Large** (Wang et al., 2022a), **BGE-base** and **BGE-Large** (Xiao et al., 2023), as well as **OpenAI-Embedding-v3-Small**, all of which use supervision from multiple tasks to enhance generalization across diverse domains.

4. For instruction-tuned dense retrieval models, which aim to align the retriever with human instructions, we include **E5-Mistral-7B-instruct** (Wang et al., 2023a), and **GritLM-7B** (Muennighoff et al., 2024), which are trained on large-scale, diverse instruction datasets to follow task-specific intents effectively.

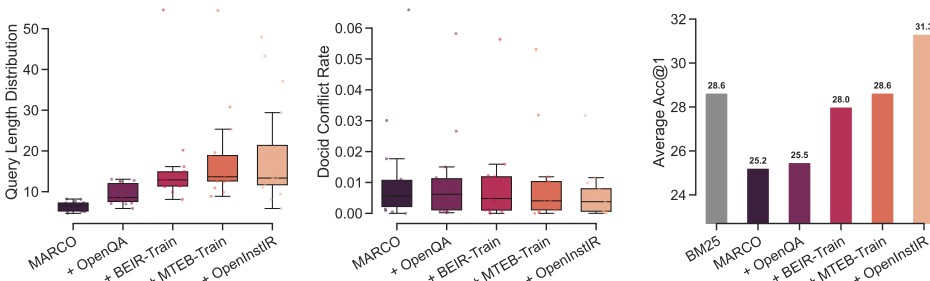

Figure 2: Model performance on unseen-dev tasks as a function of the number of training tasks. We increase the number of training tasks, starting from MS MARCO, and incrementally add open-domain QA datasets (e.g., NQ), BEIR-Train sets (e.g., NFC), MTEB-Train data (e.g., NLI), and finally the OpenInstIR collection, which includes 60 tasks across 6 domains. (Left): More instruction-tuning tasks lead to more diverse queries. (Middle): More instruction-tuning tasks reduce docid conflicts. (Right): More instruction-tuning tasks improve the Acc@1 score.

## 6 ABLATION STUDY

We conduct a systematic study on the development set, analyzing how task diversity in instruction tuning (Section 6.1), docid design (Section 6.2) and distribution (Section 6.3), corpus indexing strategy, decoding strategy (Section 6.4), and model size (Section 6.5) affect generative retrieval performance.

### 6.1 ANALYSIS OF TASK DIVERSITY

A key factor in enhancing the performance of LM-based tasks is scaling, i.e., increasing model size or data volume. The effectiveness of ZEROGR stems from instruction fine-tuning on multi-task IR datasets, which improves the instruction-following abilities of both the query generator and the title generator models. To investigate the impact of multi-task training, we curate training data with varying numbers of tasks: (a) *MS MARCO*, which contains a single task (i.e., MS MARCO (Campos et al., 2016)) and is commonly used in previous GR work; (b) + *OpenQA*, which adds popular open-domain question answering datasets, including NQ (Kwiatkowski et al., 2019) and HotpotQA; (c) + *BEIR-Train*, which incorporates the training splits of BEIR (Thakur et al., 2021), such as NFCorpus and Quora; (d) + *MTEB-Train*, which includes additional tasks from MTEB (Muennighoff et al., 2022) that are not covered in BEIR, such as NLI (we use the public BGE training split to collect these data); and (e) + *OpenInstIR*, which includes the data we collected from the training split of the MAIR (Sun et al., 2024) task collection, comprising 69 tasks from 6 domains (Figure 2).

Figure 2 shows the evaluation results of models (both query generator and docid generator) trained with different levels of task diversity, evaluated on the unseen task subset (i.e., tasks not included in any training set) of MAIR. The left plot in Figure 2 shows the distribution of average query length across tasks. We observe that models trained on more IR tasks generate queries with greater length diversity, indicating task-aware query generation strategies. In contrast, the baseline model trained only on MS MARCO produces short queries, averaging 8 words. The middle plot shows the docid conflict rate, i.e., the percentage of documents in the corpus assigned the same docid by the docid generator. Models trained on diverse tasks exhibit lower conflict rates, suggesting a stronger ability to process heterogeneous corpora. The MS MARCO baseline shows higher conflict on several diverse tasks. Finally, the right plot reports retrieval performance (top-1 accuracy) for different models. We observe consistent performance improvements on unseen tasks as training data diversity increases.

### 6.2 ANALYSIS OF DIFFERENT DOCID DESIGNS

Figure 3 compares our proposed unified docid with previous GR docid designs, while keeping all other factors (e.g., query generator, model choice, optimization strategy) constant to ensure an apple-to-apple comparison of docid effectiveness. The compared docid designs include: (i) **Random** (Tay et al., 2022), a baseline that assigns each document a random string as its docid; (ii) **Sentence** (Bevilacqua et al., 2022), which uses all sentences of each document as its docid; (iii)

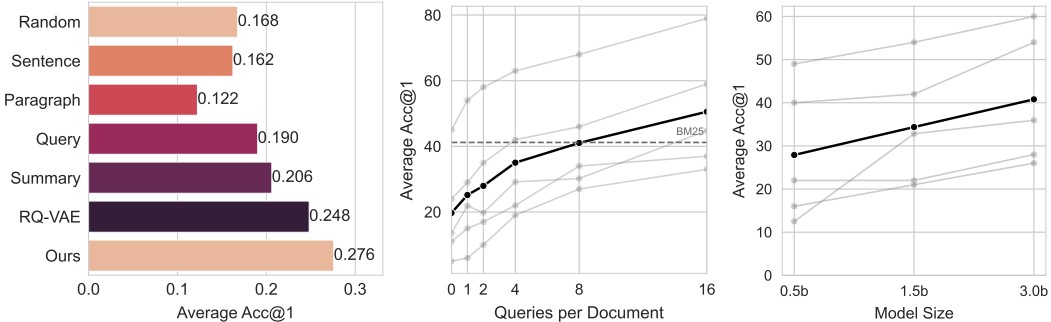

Figure 3: (Left): Comparison of different docid designs. (Middle): Acc@1 vs. generated queries per document. (Right): Acc@1 vs. model size. Gray curves are per-task scores.

**Paragraph** (Tay et al., 2022), which takes the first paragraph of each document as its docid; (iv) **Query** (Tang et al., 2023), which uses a query generator to produce a single query per document as its docid; (v) **Summary**, as introduced in (Li et al., 2024), which uses the output of a summarization model as the docid; and (vi) **RQ-VAE** (Zeng et al., 2023), which trains a RQ-VAE model on document embeddings produced by the `BGE-Large` model, enabling quantization of document embeddings into a sequence of tokens. This is a widely adopted docid representation in competitive GR systems.

### 6.3 ANALYSIS OF DOCID DISTRIBUTION

Figure 4 (left) evaluates conflict rates when treating only a prefix of the generated docid as the identifier. The results show that the conflict rate drops below 1% once the prefix length exceeds six words, and with an eight-word prefix the conflict rate is only 0.45%. This confirms that our default docid length is sufficient and that retrieval accuracy is not meaningfully impacted by docid length.

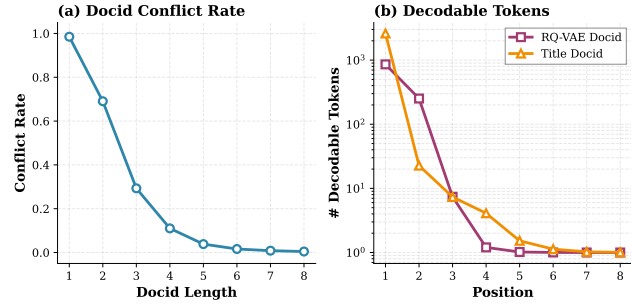

Figure 4 (right) shows the average number of decodable tokens at each trie step for RQVAE and title-based

Figure 4: (Left): Docid conflict rate wrt docid length. (Right): Average number of decodable tokens at each position, for RQ-VAE docid and our title docid.

docids. Title-based docids yield a large branching factor at the first step, with many decodable tokens, followed by a sharp reduction in later steps. In contrast, RQVAE forms a more gradually narrowing trie that becomes nearly deterministic after two to three steps.

### 6.4 ANALYSIS OF DECODING STRATEGIES

In Figure 5, we compare our reverse annealing decoding with other popular decoding algorithms, including greedy decoding (i.e., greedily sampling from the GR model without replacement), nucleus sampling with a top-p of 0.9, and beam search. All methods decode the top-100 docids for evaluation. From the results, we observe that greedy decoding achieves the best performance in terms of Acc@1, but lacks diversity and yields low recall. Nucleus sampling performs poorly on Acc@1 but achieves

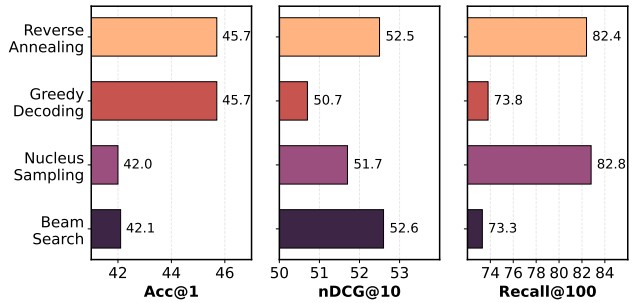

Figure 5: Ablation study of decoding algorithms across different metrics. Our proposed reverse annealing decoding achieves a good balance between precision and recall.

Table 2: Combined domain-wise results on MAIR (Acc@1) and BEIR (nDCG@10). Performance of different retrieval models across various domains. See Table 8 and 9 for details.

| Model | MAIR (38 Tasks) | | | | | | | BEIR (11 Tasks) | | | | |
|---|---|---|---|---|---|---|---|---|---|---|---|---|
| | Avg | Web. | Aca. | Legal | Med. | Fin. | Cod. | Avg | Web. | Aca. | Med. | Fin. |
| BM25 | 36.1 | 34.3 | 39.2 | 34.5 | 42.4 | 40.0 | 17.3 | 42.4 | 45.4 | 38.8 | 32.7 | 41.6 |
| Contriever | 33.6 | 39.8 | 33.4 | 26.8 | 30.8 | 37.3 | 17.7 | 47.6 | 51.5 | 43.0 | 33.9 | 47.6 |
| GTR-T5-base | 32.5 | 36.0 | 33.6 | 25.3 | 31.9 | 37.4 | 18.7 | 45.3 | 50.7 | 35.3 | 32.7 | 45.3 |
| GTR-T5-large | 35.4 | 39.8 | 39.6 | 27.8 | 31.8 | 38.5 | 24.0 | 48.0 | 53.3 | 37.4 | 33.4 | 50.0 |
| E5-Base | 37.2 | 36.2 | 48.6 | 28.5 | 35.3 | 44.9 | 26.7 | 48.9 | 51.8 | 46.1 | 35.0 | 50.2 |
| E5-Large | 38.2 | 38.6 | 51.0 | 25.0 | 35.6 | 46.6 | 25.7 | 49.2 | 51.7 | 47.9 | 37.4 | 48.8 |
| BGE-Base | 37.0 | 38.6 | 40.2 | 25.8 | 37.6 | 42.2 | 29.0 | 50.5 | 52.5 | 47.1 | 36.0 | 55.2 |
| BGE-Large | 39.4 | 39.4 | 46.2 | 36.0 | 37.2 | 45.1 | 29.0 | 51.8 | 53.8 | 47.9 | 38.1 | 56.5 |
| OpenAI-Embed | 40.6 | 40.6 | 48.2 | 31.0 | 39.7 | 49.4 | 28.7 | 54.2 | 56.3 | 47.2 | 37.6 | 63.4 |
| E5-mistral-7B | 46.8 | 45.4 | 55.4 | 42.3 | 43.1 | 55.3 | 40.0 | 55.7 | 56.4 | 48.6 | 39.6 | 68.8 |
| GritLM-7B | 47.0 | 44.1 | 58.2 | 43.3 | 42.6 | 57.6 | 40.0 | 45.0 | 47.7 | 48.2 | 36.9 | 37.8 |
| ZeroGR-3B | 41.1 | 42.7 | 47.4 | 40.0 | 38.3 | 39.2 | 36.3 | 48.1 | 49.2 | 45.8 | 34.7 | 53.8 |

Table 3: Performance of different generative retrieval models across various datasets on BEIR.

| Method | Training data | Avg | Argu. | SciF. | NFC. | FiQA | SciD. | Covid |
|---|---|---|---|---|---|---|---|---|
| GENRE (Cao et al., 2020) | GPL | 23.0 | **42.5** | 42.3 | 20.0 | 11.6 | 6.8 | 14.7 |
| GENRET (Sun et al., 2023a) | GPL | 41.1 | 34.3 | 63.9 | 31.6 | 30.2 | 14.9 | 71.8 |
| GLEN (Lee et al., 2023) | NQ320k | – | 17.6 | – | 15.9 | – | – | – |
| TIGER (Rajput et al., 2023) | OpenInstIR | 31.0 | 14.0 | 37.0 | **39.5** | 16.0 | 14.0 | 65.7 |
| ZeroGR (Ours) | OpenInstIR | **44.9** | 35.4 | **72.8** | 34.7 | **34.1** | **18.7** | **73.5** |

high recall. In contrast, reverse annealing strikes a good balance between precision and recall, achieving competitive results across all metrics.

## 6.5 Analysis of query number and model size

The middle section of Figure 3 illustrates the impact of the number of queries generated per document on the average top-1 accuracy of ZEROGR. We observe a clear upward trend: as the number of queries increases, the retrieval performance improves steadily. This highlights the importance of diverse query views for better semantic coverage during indexing. Notably, when using eight queries per document, ZEROGR already reaches performance on par with the strong sparse baseline BM25. Further increasing the query count to sixteen enables ZEROGR to surpass BM25, suggesting that high query diversity provides richer signals for matching user queries to relevant documents.

The right section of Figure 3 examines how the size of the backbone language model affects retrieval performance. For this analysis, we adopt a series of Qwen2.5 (Qwen et al., 2025) models with varying parameter scales. The results demonstrate a consistent gain in top-1 accuracy on unseen IR tasks as the model size grows, implying that larger models benefit from enhanced generalization and better understanding of the instruction-based retrieval formulation. This finding underscores the value of scaling up model capacity in generative retrieval frameworks, particularly in zero-shot settings.

## 7 Benchmark Evaluation

### 7.1 Evaluation results on MAIR

As shown in Table 2 (MAIR), our proposed ZEROGR framework demonstrates strong performance across a wide range of retrieval tasks. It achieves an average score of 41.1 (Acc@1), substantially outperforming traditional sparse retrieval methods like BM25 and widely adopted dense retrieval models such Contriever, GTR, E5, BGE, and on par with instruction-tuned OpenAI-Embedding-v3-Small. These results highlight the effectiveness of our instruction-based generative retrieval approach

in capturing deeper semantic relevance. Our detailed experimental results on MAIR are shown in Table 8 (Appendix).

The performance gains of ZEROGR are not limited to familiar tasks but also generalize well to unseen domains. Notably, the model performs better than baselines on several previously unseen datasets, including Apple, MB, PM.A, DD, and NCL (see Table 8, Appendix). This demonstrates the robustness and transferability of the approach, as it adapts effectively to new retrieval settings without requiring additional task-specific supervised data.

Using a 3B LLM, ZEROGR can achieve strong performance across different tasks compared to baselines, though it still underperforms large embedding models such as GritLM-7B and E5-Mistral-7B. This indicates that our design is highly parameter-efficient, achieving strong performance across diverse tasks without relying on massive model scaling.

## 7.2 EVALUATION RESULTS ON BEIR

As shown in Table 2 (BEIR), ZEROGR outperforms several baselines such as BM25, Contriever, GTR, and GritLM-7B, but still underperforms other dense retrieval methods. We have more detailed comparison between ZeroGR and dense retrieval on BEIR in Appendix C. Furthermore, Table 3 compares ZEROGR with previous generative retrieval baselines on BEIR, which we can see our method achieves best performance among most datasets.

## 8 CONCLUSION

We have presented ZEROGR, an instruction-driven framework that extends generative retrieval to zero-shot scenarios. By unifying three key components, viz. a model-based docid generator, an instruction-conditioned query generator, and a reverse-annealed decoding algorithm, ZEROGR transforms a corpus and a natural-language task description into a task-specific generative index without requiring supervision. Systematic ablation studies along task diversity, query volume, and model size reveal consistent performance improvements. Empirical evaluations on MAIR tasks and BEIR datasets demonstrate the effectiveness of ZEROGR.

The limitations of this work include a lack of evaluation on large-scale corpora (e.g., those with over 1M documents) and the use of relatively small LLMs (our largest model is only 3B). We believe future work is required to scale both the corpus size and the model size.

### ACKNOWLEDGEMENTS

We thank our reviewers for their helpful feedback. This research was (partially) supported by the Dutch Research Council (NWO), under project numbers 024.004.022, NWA.1389.20.183, and KICH3.LTP.20.006, and the European Union under grant agreements No. 101070212 (FINDHR) and No. 101201510 (UNITE). Views and opinions expressed are those of the author(s) only and do not necessarily reflect those of their respective employers, funders and/or granting authorities.

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

# A   OpenInstIR

We collect the training set OpenInstIR based on the training splits of published information retrieval datasets (e.g., these from (Adlakha et al., 2021; Austin et al., 2021; Boteva et al., 2016; Chalkidis et al., 2021; Chen et al., 2022b;b;b;c; Cohan et al., 2020; Dinan et al., 2018; Hendrycks et al., 2021a;b;c; Hoffart et al., 2011; Husain et al., 2019; Kornilova and Eidelman, 2019; Levy et al., 2017; Li et al., 2023a; Liu et al., 2023; Maia et al., 2018; Muennighoff et al., 2023; Welleck et al., 2021; Zhang et al., 2025a; Zhu et al., 2022)). Specifically, we pair each query with its corresponding positive document and annotate the pairs with natural language instructions following MAIR, forming standardized (instruction, query, document) triples entirely from human annotations. Table 4 reports statistics of OpenInstIR, which spans 6 domains and 66 tasks; during training, we upsample tasks with fewer than 5K samples and downsample those with more than 500K samples. Table 5 compares query and document types between OpenInstIR and the evaluation set, showing 9 shared query types and 14 shared document types, while the test set contains 9 additional query types and 8 additional document types unseen during training.

Table 4: OpenInstIR dataset statistics by domain.

| Dataset | Size | Dataset | Size | Dataset | Size | **Dataset** | Size |
|---|---|---|---|---|---|---|---|
| *Academic* | | | | | | | |
| S2-TC | 100K | S2-AC | 100K | S2-TA | 100K | TAD | 208K |
| TAS2 | 108K | StackMath | 47K | Proof-P | 16K | Proof-R | 2K |
| Stacks-P | 11K | Stacks-R | 9K | CompMath | 8K | SciDocs | 1K |
| SciFact | 809 | LitSearch | 146 | | | | |
| *Code* | | | | | | | |
| CSN | 1.88M | CodeEdit | 21K | SWE-B | 19K | RepoBench | 17K |
| HF-API | 8K | TLDR | 6K | TensorAPI | 6K | APPS | 5K |
| LeetCode | 2K | CoNaLa | 2K | TorchAPI | 837 | HumanEval-X | 720 |
| MBPP | 374 | | | | | | |
| *Finance* | | | | | | | |
| USNews | 10K | FinQA | 6K | FiQA | 6K | HC3-Fin | 3K |
| ConvFinQA | 3K | Goldman | 2K | TAT-DQA | 1K | TradeEvt | 900 |
| *Legal* | | | | | | | |
| LePaRD | 22.7M | CLERC | 327K | BillSum | 19K | REGIR-UK2EU | 2K |
| REGIR-EU2UK | 2K | BSARD | 886 | CUAD | 717 | | |
| *Medical* | | | | | | | |
| PMQA-C | 197K | PMQA-A | 197K | Huatuo | 25K | NFCorpus | 3K |
| CARE | 77 | | | | | | |
| *Web* | | | | | | | |
| Reddit | 12.7M | AGNews | 1.16M | CC-News | 708K | XSum | 204K |
| zsRE | 148K | Fever | 110K | ToT | 109K | WoW | 64K |
| TopiOCQA | 45K | AY2 | 18K | CQADup | 13K | InstructIR | 10K |
| Quora | 10K | WnCw | 5K | TREx | 5K | ExcluIR | 3K |
| NevIR | 2K | ArguAna | 1K | | | | |

# B   Ablation Study Details

To reduce computational overhead, our ablation studies are conducted on the Dev subset rather than the full test set; Table 6 provides the task breakdown. For the docid design ablation (Section 6.2), Table 7 presents representative docids, from which we observe that sentence, paragraph, and IDF docids are often noisy; query and summary docids fail to capture key concepts early and are suboptimal for left-to-right GR decoding; and RQ-VAE does not exploit the language generation capabilities of LLMs. For the decoding ablation study, we fix the beam size to 100 and use the same hyperparameters as in the main experiments. No top-k or top-p pruning is applied, and the reverse annealing parameter is kept consistent across all datasets and models.

Table 5: Query and doc types between our train and eval set.

| Query types | Document types |
|---|---|
| *Training ∩ Evaluation* | |
| Question, Dialog, Claim, Function Header, NL Command, Code Problem, Math question, Paper Title, Summary | Document, Answer, Function, Command Doc, Solution, Article, Articles, Medical Doc, Paragraph, Page, Statute, Table |
| *(9 types)* | *(14 types)* |
| *Only in Evaluation* | |
| Health Record, Topic, Situation, Request, Patient Data & Description, Medical Case, Medical Claim, Numerical Claim | Clinical Trials, Prior Case, Communications, Dataset, Music, Tweet, News, POI, Table |
| *(9 types)* | *(8 types)* |
| *Only in Training* | |
| Math Statement, Entity & Relation, Paper Abstract, Entity Mention, CNL Command, GitHub Issue, Commit, Code Context, Math Question, Title, EU Directive, UK Legislation, Instruction, Reaction, Description | Entity Page, Citation, Proof, Reference, Duplicate Question, Related File, Code Diff, Next Function, HuggingFace API, Tensor API, PyTorch API, UK Legislation, EU Directive, Highlight, Proteins Documents, Wikipedia Page |
| *(14 types)* | *(16 types)* |

Table 6: Development set for ablation study.

| Name | Model | Task List (with query count) |
|---|---|---|
| Figure 2, Figure 3 (left) | Llama-1B | {ToolBench (100), AILA2019-Case (50), NFCorpus (100), SciFact (100), ArguAna (100), LitSearch (100), ClinicalTrials_2023 (37), FinanceBench (100), SciDocs (100), News21 (100), TopiOCQA (100), Touche (49), FiQA (100)} |
| Figure 3 (middle, right) | Llama-1B, or Qwen2.5 | {LeetCode (100), Competition-Math (100), TMDB (100), Stein_Proof (64), PytorchAPI (100)} |
| Figure 5 | Llama-1B | {Leetcode (100), Math (100), BillSum (100), SciFact (100), TAT-DQA (70), ConvFinQA (96)} |

## C  BENCHMARK RESULTS

On BEIR, our method underperforms state-of-the-art dense retrieval models (Tables 8 and 9). We attribute this gap to two main factors. First, many BEIR tasks overlap with the large-scale training data used by modern embedding models, whereas MAIR is a newer and more diverse benchmark on which these models generalize less effectively, giving dense retrieval an inherent advantage on BEIR. Second, dense retrieval has benefited from years of targeted optimization for zero-shot evaluation, including hard-negative mining, distillation, and large-scale pre-training Wang et al. (2022a); Xiong et al. (2020). In contrast, our work represents one of the first efforts to systematically improve zero-shot generative retrieval across heterogeneous tasks using a simple training objective (Eq. 4), making a performance gap with mature dense systems expected. Nevertheless, our approach substantially narrows this gap compared to prior generative retrieval methods. Notably, while our query generator is trained on diverse data, the retrieval model itself is initialized from Llama-3B-Instruct and trained solely on synthetic data, without supervision from IR datasets; incorporating IR-specific pre-training or advanced techniques such as hard-negative mining and distillation may further reduce the remaining gap.

## D  MODEL EFFICIENCY

Table 10 reports the estimated computational cost of dense retrieval and generative retrieval methods (Li et al., 2024; Rajput et al., 2023; Wang et al., 2023a) under a fixed setting (Llama-3B, single A100 GPU with vLLM) on a 100K-document corpus for top-($K$=1) retrieval (noting that GR slows

Table 7: Examples of different types of docids.

| Type | Example |
|---|---|
| Random | asd8xc2c9ma90xj2398 |
| Sentence | LIMASSOL, Cyprus, April 28, 2021 /PRNewswire/ – One of the top financial investment firms of the FX industry, Windsor Brokers .... |
| Paragraph | LIMASSOL, Cyprus, April 28, 2021 /PRNewswire/ – One of the top financial investment firms of the FX industry, Windsor Brokers .... |
| Query | Induction of myelodysplasia by myeloid-derived suppressor cells. |
| Summary | 1. Game of Thrones season 7 2. Plot and storyline 3. New cast members 4. Filming locations 5. Critical reception and ratings |
| RQ-VAE | `<|g16289|> <|g13509|> <|g10485|> <|g11274|> <|g369|> <|g3661|> <|g13026|> <|g8187|>` |
| IDF | brokerswindsor mt4 brokerswere kontos windsorbrokers |
| Ours | rna folding computational methods thermodynamic optimization |

Table 8: Model performance (top-1 retrieval accuracy) on seen and unseen subset of MAIR.

| Dataset | | | | | | | Seen Subset | | | | | | | | | | Unseen Subset | | | |
|---|---|---|---|---|---|---|---|---|---|---|---|---|---|---|---|---|---|---|---|---|
| Model | Avg | FiQA | NFC.s | SciD. | SciF. | ToQA. | TAT | CoF. | LeetC. | LitSe. | BiSum | CodeSe. | Math | ConvF. | Conala | StMath | Apple | FinBen | AILAC | AILAS |
| BM25 | 36.1 | 24.0 | 45.5 | 16.0 | 53.0 | 11.0 | 67.1 | 47.9 | 12.0 | 66.0 | 69.0 | 33.0 | 41.0 | 47.9 | 7.0 | 20.0 | 52.1 | 9.0 | 14.0 | 10.0 |
| Contriever | 33.6 | 33.0 | 43.5 | 17.0 | 62.0 | 11.0 | 54.3 | 42.7 | 7.0 | 39.0 | 54.0 | 37.0 | 36.0 | 42.7 | 9.0 | 13.0 | 50.7 | 6.0 | 12.0 | 6.0 |
| GTR-T5-base | 32.5 | 33.0 | 43.0 | 12.0 | 50.0 | 16.0 | 58.6 | 37.5 | 6.0 | 41.0 | 47.0 | 41.0 | 49.0 | 37.5 | 9.0 | 16.0 | 47.9 | 10.0 | 4.0 | 10.0 |
| GTR-T5-large | 35.4 | 45.0 | 43.5 | 14.0 | 55.0 | 12.0 | 40.0 | 42.7 | 10.0 | 43.0 | 59.0 | 51.0 | 63.0 | 42.7 | 11.0 | 23.0 | 50.7 | 14.0 | 6.0 | 6.0 |
| E5-Base | 37.2 | 41.0 | 40.0 | 17.0 | 63.0 | 16.0 | 61.4 | 52.1 | 11.0 | 49.0 | 61.0 | 59.0 | 78.0 | 52.1 | 10.0 | 36.0 | 52.1 | 18.0 | 6.0 | 12.0 |
| E5-Large | 38.2 | 45.0 | 45.5 | 20.0 | 67.0 | 13.0 | 70.0 | 57.3 | 9.0 | 49.0 | 52.0 | 57.0 | 75.0 | 57.3 | 13.0 | 44.0 | 47.9 | 13.0 | 12.0 | 6.0 |
| BGE-Base | 37.0 | 43.0 | 43.5 | 20.0 | 62.0 | 13.0 | 58.6 | 41.7 | 10.0 | 44.0 | 67.0 | 64.0 | 50.0 | 41.7 | 13.0 | 25.0 | 47.9 | 20.0 | 8.0 | 8.0 |
| BGE-Large | 39.4 | 51.0 | 46.5 | 22.0 | 65.0 | 13.0 | 60.0 | 44.8 | 13.0 | 56.0 | 68.0 | 66.0 | 66.0 | 44.8 | 8.0 | 22.0 | 46.6 | 23.0 | 8.0 | 8.0 |
| OpenAI-Embed | 40.6 | 51.0 | 51.0 | 22.0 | 60.0 | 19.0 | 62.9 | 51.0 | 6.0 | 53.0 | 59.0 | 67.0 | 73.0 | 51.0 | 13.0 | 33.0 | 52.1 | 30.0 | 10.0 | 10.0 |
| GTE-Qwen2-1.5B | 44.4 | 54.0 | 50.0 | 24.0 | 69.0 | 25.0 | 65.7 | 65.6 | 41.0 | 63.0 | 79.0 | 70.0 | 84.0 | 65.6 | 20.0 | 40.0 | 47.9 | 33.0 | 10.0 | 10.0 |
| E5-mistral-7B | 46.8 | 60.0 | 50.5 | 17.0 | 67.0 | 14.0 | 67.1 | 64.6 | 36.0 | 68.0 | 74.0 | 54.0 | 78.0 | 64.6 | 30.0 | 47.0 | 43.8 | 41.0 | 12.0 | 38.0 |
| GritLM-7B | 47.0 | 63.0 | 49.5 | 29.0 | 69.0 | 17.0 | 85.7 | 62.5 | 46.0 | 60.0 | 74.0 | 53.0 | 87.0 | 62.5 | 21.0 | 46.0 | 43.8 | 33.0 | 12.0 | 42.0 |
| **ZeroGR-3B** | 41.1 | 37.0 | 36.5 | 24.0 | 51.0 | 13.0 | 38.6 | 57.3 | 36.0 | 41.0 | 81.0 | 61.0 | 81.0 | 57.3 | 12.0 | 40.0 | 52.1 | 11.0 | 12.0 | 22.0 |

| | | | | | | | | Unseen subset | | | | | | | | | | | | |
|---|---|---|---|---|---|---|---|---|---|---|---|---|---|---|---|---|---|---|---|---|
| | ACOR. | CPCD | CORE | MB. | PM. | PM.A | CliDS | CliT23 | DD | Table | QuanT | PoRec | Monant | NCL. | NCL.T | Legal | Geno. | Touche | CliT21 | News21 |
| BM25 | 32.8 | 1.0 | 37.5 | 83.8 | 53.9 | 6.5 | 28.3 | 51.4 | 15.6 | 10.0 | 86.9 | 24.5 | 67.4 | 50.7 | 22.2 | 45.0 | 52.8 | 59.2 | 33.3 | 10.9 |
| Contriever | 40.4 | 1.0 | 52.5 | 89.2 | 32.9 | 0.0 | 6.7 | 37.8 | 21.3 | 8.3 | 76.8 | 46.7 | 65.0 | 60.0 | 34.2 | 35.0 | 27.8 | 52.0 | 32.7 | 23.8 |
| GTR-T5-base | 31.3 | 1.0 | 47.5 | 89.2 | 36.8 | 1.6 | 13.3 | 36.5 | 13.6 | 12.5 | 77.8 | 37.7 | 70.0 | 48.0 | 22.2 | 40.0 | 25.0 | 55.1 | 29.3 | 15.6 |
| GTR-T5-large | 34.8 | 3.0 | 60.0 | 91.9 | 31.6 | 1.6 | 8.3 | 39.2 | 16.2 | 10.0 | 78.8 | 48.7 | 68.0 | 53.3 | 23.9 | 40.0 | 25.0 | 65.3 | 37.3 | 19.1 |
| E5-Base | 40.4 | 3.0 | 42.5 | 81.1 | 43.4 | 6.5 | 11.7 | 39.2 | 13.2 | 5.8 | 78.8 | 54.2 | 72.0 | 55.3 | 27.4 | 35.0 | 33.3 | 37.8 | 36.0 | 14.5 |
| E5-Large | 38.4 | 3.0 | 45.0 | 86.5 | 36.8 | 4.8 | 15.0 | 40.5 | 12.3 | 7.5 | 80.8 | 52.5 | 71.0 | 55.3 | 47.9 | 30.0 | 38.9 | 41.8 | 32.0 | 17.2 |
| BGE-Base | 39.4 | 0.0 | 45.0 | 91.9 | 48.7 | 0.0 | 30.0 | 31.1 | 16.4 | 8.3 | 81.8 | 44.4 | 70.0 | 57.3 | 42.7 | 20.0 | 36.1 | 41.8 | 41.3 | 19.9 |
| BGE-Large | 36.9 | 0.0 | 52.5 | 94.6 | 42.1 | 3.2 | 23.3 | 28.4 | 17.8 | 5.0 | 81.8 | 50.4 | 74.0 | 54.0 | 40.2 | 60.0 | 38.9 | 51.0 | 41.3 | 15.2 |
| OpenAI-Embed | 32.8 | 1.0 | 55.0 | 86.5 | 46.1 | 3.2 | 25.0 | 44.6 | 13.5 | 12.5 | 86.9 | 47.3 | 76.0 | 57.3 | 49.6 | 45.0 | 33.3 | 52.0 | 38.0 | 13.7 |
| GTE-Qwen2-1.5B | 37.9 | 5.1 | 70.0 | 81.1 | 14.5 | 8.1 | 20.0 | 17.6 | 14.8 | 14.2 | 85.9 | 58.6 | 74.2 | 61.3 | 51.3 | 40.0 | 61.1 | 65.3 | 31.3 | 23.0 |
| E5-mistral-7B | 41.9 | 5.0 | 60.0 | 83.8 | 43.4 | 1.6 | 46.7 | 48.6 | 19.4 | 11.7 | 83.8 | 66.1 | 71.0 | 62.7 | 58.1 | 45.0 | 36.1 | 58.2 | 46.7 | 25.4 |
| GritLM-7B | 35.4 | 7.0 | 65.0 | 70.3 | 59.2 | 0.0 | 28.3 | 45.9 | 14.8 | 10.8 | 86.9 | 72.2 | 77.0 | 63.3 | 50.4 | 45.0 | 33.3 | 57.1 | 47.3 | 22.7 |
| **ZeroGR-3B** | 26.7 | 4.0 | 55.0 | 89.2 | 23.9 | 12.9 | 25.0 | 37.9 | 44.4 | 12.0 | 79.8 | 60.9 | 69.7 | 65.3 | 36.2 | 45.0 | 58.3 | 46.9 | 42.0 | 21.5 |

with larger $K$, while Dense slows with larger corpora). ZeroGR is more efficient than prior GR methods due to its compact docid design and reduced training epochs. Compared with dense retrieval, GR incurs a much higher one-time indexing cost due to model training, which can be amortized during inference.

# E PROMPTS

```
1. **Length**: Strictly 6-8 words (terms/words)
2. **Term Inclusion**: Must include 3-5 core terms directly from
the document
3. **Term Positioning**: Rank by relevance and importance (highest
→ lowest, general → specific)
```

Table 9: nDCG@10 on BEIR benchmark datasets.

| Category | Method | Avg. | ArguAna | SciFact | NFCorpus | FiQA | SciDocs |
|---|---|---|---|---|---|---|---|
| Sparse | BM25 | 42.3 | 32.7 | 65.1 | 32.7 | 24.8 | 12.4 |
| DR | Contriever | 47.6 | 32.1 | 70.3 | 33.9 | 35.5 | 15.7 |
| DR | GTR-T5-base | 45.3 | 32.7 | 58.6 | 32.7 | 34.5 | 12.1 |
| DR | GTR-T5-large | 48.0 | 34.3 | 61.9 | 33.4 | 43.3 | 12.8 |
| DR | E5-Base | 48.9 | 31.1 | 73.9 | 35.0 | 39.6 | 18.3 |
| DR | E5-Large | 49.2 | 31.7 | 76.3 | 37.4 | 42.3 | 19.6 |
| DR | BGE-Base | 50.5 | 41.8 | 74.3 | 36.0 | 43.4 | 19.8 |
| DR | BGE-Large | 51.8 | 41.6 | 75.2 | 38.1 | 48.5 | 20.6 |
| DR | E5-mistral-7B | 55.7 | 44.1 | 76.6 | 39.6 | 59.7 | 20.7 |
| DR | GritLM-7B | 45.0 | 40.7 | 76.8 | 36.9 | 44.1 | 19.6 |
| DR | OpenAI Embed | 54.2 | 37.1 | 73.1 | 37.6 | 48.5 | 21.2 |
| GR | GENRE | – | 42.5 | 42.3 | 20.0 | 11.6 | 6.8 |
| GR | GENRET | – | 34.3 | 63.9 | 31.6 | 30.2 | 14.9 |
| GR | GLEN | – | 17.6 | – | 15.9 | – | – |
| GR | TIGER (Llama-3B) | – | 14.0 | 37.0 | 39.5 | 16.0 | 14.0 |
| GR | ZeroGR-3B | 48.1 | 35.4 | 72.8 | 34.7 | 34.1 | 18.7 |

| Category | Method | Touche | TREC-News | Fever | Quora | Covid | CQADupStack |
|---|---|---|---|---|---|---|---|
| Sparse | BM25 | 59.0 | 20.7 | 58.3 | 73.8 | 58.3 | 28.0 |
| DR | Contriever | 42.5 | 27.3 | 90.6 | 86.6 | 59.6 | 29.9 |
| DR | GTR-T5-base | 48.1 | 22.5 | 83.2 | 88.7 | 56.1 | 28.9 |
| DR | GTR-T5-large | 53.1 | 26.6 | 86.8 | 89.1 | 56.7 | 29.8 |
| DR | E5-Base | 41.1 | 22.9 | 91.1 | 86.4 | 60.7 | 38.3 |
| DR | E5-Large | 34.8 | 25.3 | 93.1 | 86.9 | 55.2 | 38.5 |
| DR | BGE-Base | 41.4 | 21.2 | 85.6 | 89.8 | 67.1 | 35.1 |
| DR | BGE-Large | 45.5 | 21.4 | 86.6 | 89.3 | 64.5 | 38.3 |
| DR | E5-mistral-7B | 46.8 | 29.4 | 91.8 | 84.8 | 77.8 | 41.4 |
| DR | GritLM-7B | 21.5 | 34.9 | 68.9 | 84.9 | 31.5 | 35.0 |
| DR | OpenAI Embed | 47.5 | 26.2 | 92.8 | 89.9 | 78.2 | 44.1 |
| GR | GENRE | – | – | – | – | 14.7 | – |
| GR | GENRET | – | – | – | – | 71.8 | – |
| GR | GLEN | – | – | – | – | – | – |
| GR | TIGER (Llama-3B) | 58.1 | 16.4 | – | 59.6 | 65.7 | – |
| GR | ZeroGR-3B | 37.5 | 23.5 | 86.7 | 76.7 | 73.5 | 35.2 |

Table 10: Estimated indexing and retrieval cost.

| Method | Indexing (offline) | | Retrieval (online per query) | |
|---|---|---|---|---|
| | FLOPs | Est. time | FLOPs | Latency |
| Dense-3B (Wang et al., 2023a) | $6.00 \times 10^{17}$ | ∼8.0 min | $3.48 \times 10^{12}$ | ∼2.8 ms |
| GENRET-3B (Sun et al., 2023a) | $4.60 \times 10^{19}$ | ∼10.2 h | $1.34 \times 10^{12}$ | ∼1.1 ms |
| Summary-3B (Li et al., 2024) | $1.15 \times 10^{19}$ | ∼2.6 h | $4.80 \times 10^{12}$ | ∼3.8 ms |
| ZeroGR-3B | $8.06 \times 10^{18}$ | ∼1.8 h | $1.34 \times 10^{12}$ | ∼1.1 ms |

```
4. **Formatting**:
   - Use lowercase letters, numbers, and spaces only
   - Preserve special terms/symbols (e.g., PD3.1)
   - **No articles** (a, the), **linking verbs**, or auxiliary
verbs
   - **No verbs** (use nouns/adjectives only)
5. **Requirements**:
   - Terms must be derivable from the document
   - Ensure uniqueness and precise core content representation
```

Table 11: Dense retrieval model information.

| Model | Size | Training Data | Link |
|---|---|---|---|
| BM25 | N/A | N/A | https://github.com/cvangysel/BM25S |
| Contriever-MARCO | 110M | MS MARCO | https://github.com/facebookresearch/contriever |
| GTR-base | 110M | MS MARCO | https://huggingface.co/google/gtr-base |
| GTR-large | 335M | MS MARCO | https://huggingface.co/google/gtr-large |
| E5-base | 110M | unknown | https://huggingface.co/intfloat/e5-base-v2 |
| E5-large | 335M | unknown | https://huggingface.co/intfloat/e5-large-v2 |
| BGE-base | 110M | MTEB-Train | https://huggingface.co/BAAI/bge-base-en-v1.5 |
| BGE-large | 335M | MTEB-Train | https://huggingface.co/BAAI/bge-large-en-v1.5 |
| OpenAI-Embed-Small | unknown | unknown | https://platform.openai.com/docs/guides/embeddings |
| E5-Mistral-7B-instruct | 7B | E5 (LLM generated) | https://huggingface.co/intfloat/e5-mistral-7b-instruct |
| GritLM-7B | 7B | E5 (LLM generated) | https://huggingface.co/GritLM/GritLM-7B |

Table 12: Generative retrieval model information.

| Method | Training Data | Model Size | DocID Type | Decoding |
|---|---|---|---|---|
| GENRE (Cao et al., 2020) | GPL | T5-220M | Title | Beam Search |
| GENRET (Sun et al., 2023a) | GPL | T5-220M | RQ-VAE | Beam Search |
| GLEN (Lee et al., 2023) | NQ320k | T5-220M | Keywords | Beam Search |
| TIGER (Rajput et al., 2023) | OpenInstIR | Llama-3B | RQ-VAE | Reverse-Annealing |
| ZeroGR (Ours) | OpenInstIR | Llama-3B | Title | Reverse-Annealing |

## F  ADDITIONAL RESULTS

Table 11 lists the size, training data, and links of the baselines. Table 12 shows the size, training data, docid type, and decoding strategy of the generative retrieval models.

We conduct an ablation study in which we replace the doc2query-generated queries in the baseline TIGER (RQ-VAE) model with queries from our generator. As reported in Table 13, using our queries alone yields an overall improvement of +6.4. The gains are especially pronounced on domain-specific benchmarks, including legal (BillSum +30.0), finance (FinQA +26.7, TAT-DQA +20.0), code (LeetCode +24.0, CodeSearchNet +23.0), and scientific retrieval (Competition-Math +21.0, LitSearch +19.0). These results indicate that high-quality, diverse queries are crucial, particularly for specialized retrieval tasks.

## G  USE OF LARGE LANGUAGE MODELS

In the preparation of this manuscript, we used large language models (LLMs) to enhance the clarity and linguistic quality of our academic writing. Specifically, we employed Claude Sonnet 4 (Anthropic) and GPT-5 (OpenAI) for language polishing and refinement purposes.

Table 13: Performance comparison between doc2query and our method for the RQ-VAE docID baseline (TIGER (Rajput et al., 2023)).

| Task | doc2query | Our query generator | Diff |
|------|-----------|---------------------|------|
| AILA2019-Case | 2.00 | 2.00 | +0.00 |
| Apple | 13.70 | 5.48 | -8.22 |
| ArguAna | 12.00 | 11.00 | -1.00 |
| BillSum | 36.00 | 66.00 | **+30.00** |
| ClinicalTrials_2021 | 4.67 | 6.67 | +2.00 |
| ClinicalTrials_2023 | 1.35 | 2.70 | +1.35 |
| CodeEditSearch | 13.00 | 22.00 | +9.00 |
| CodeSearchNet | 33.00 | 56.00 | **+23.00** |
| Competition-Math | 40.00 | 61.00 | **+21.00** |
| Conala | 3.00 | 9.00 | +6.00 |
| ConvFinQA | 22.92 | 37.50 | +14.58 |
| FiQA | 7.00 | 13.00 | +6.00 |
| FinQA | 14.44 | 41.11 | **+26.67** |
| LeetCode | 6.00 | 30.00 | **+24.00** |
| LegalQuAD | 10.00 | 4.00 | -6.00 |
| LitSearch | 12.00 | 31.00 | **+19.00** |
| NFCorpus | 41.00 | 6.50 | -34.50 |
| News21 | 13.67 | 21.88 | +8.20 |
| SciDocs | 16.00 | 14.00 | -2.00 |
| SciFact | 34.00 | 42.00 | +8.00 |
| StackMathQA | 13.00 | 26.00 | +13.00 |
| TAT-DQA | 7.14 | 27.14 | **+20.00** |
| ToT_2023 | 3.00 | 0.00 | -3.00 |
| TopiOCQA | 18.00 | 8.00 | -10.00 |
| Touche | 46.94 | 39.80 | -7.14 |
| Average | **16.95** | **23.35** | **+6.40** |

---

**Algorithm 1** Reverse-Annealed DocID Generation

---

**Require:** Prefix tree $T$, decoder $f(\cdot)$, query $q$, number of docids $K$, maximum temperature $T_{\max}$
**Ensure:** Generated docids $\mathcal{Z}$
1: $\mathcal{Z} \leftarrow \emptyset$
2: **for** $i = 1, \ldots, K$ **do**
3:     Compute temperature $t_i \leftarrow g(i)$ (Eq. 5)
4:     Initialize prefix $\mathbf{x}_i \leftarrow \emptyset$
5:     **while** $\mathbf{x}_i$ is not a leaf of $T$ **do**
6:         Get logits $\boldsymbol{\ell} \leftarrow f(q, \mathbf{x}_i)$
7:         Sample $x \sim \mathrm{Softmax}(\boldsymbol{\ell}/t_i)\big|_T$
8:         $\mathbf{x}_i \leftarrow \mathbf{x}_i \cup \{x\}$
9:     **end while**
10:     Add docid $z_i$ to $\mathcal{Z}$ and remove its leaf from $T$
11: **end for**
12: **return** $\mathcal{Z}$

---

```python
def normalized_sigmoid(t, k=10, m=0.5):
    sigmoid = lambda z: 1 / (1 + np.exp(-z))
    a = sigmoid(k * (0 - m))
    b = sigmoid(k * (1 - m))
    return (sigmoid(k * (t - m)) - a) / (b - a)
```

Figure 6: Normalized sigmoid function. $t$ is the step number.

