# OpenReview forum: "ZeroGR: A Generalizable and Scalable Framework for Zero-Shot Generative Retrieval"
_ICLR.cc/2026/Conference — ICLR 2026 Poster_

### Official Review · Reviewer_Zbgo · 2025-10-29

**Soundness:** 2
**Presentation:** 3
**Contribution:** 3
**Rating:** 4
**Confidence:** 4

**Summary:**

This paper presents ZeroGR, a framework for zero-shot (in-domain, novel-task) generative retrieval. The framework includes three main components:

1) a protocol for training a small docid generator model that is finetuned on SOTA LLMs prompted to generate short, keyword-rich sentences from a document d_i that serves as its docid z_i.

2) an indexing protocol that trains the generative retriever model to generate docids z_i described in 1) from synthetic queries q_i,b generated from document d_i by a task-instructed query generator model.

3) A novel reverse-annealed constrained decoding scheme for docid generation at retrieval time that increases decoding temperature at each token of the docid to better balance precision (low temp, earlier tokens in docid) with recall (high temp, later tokens in docid).

They present SOTA generative retrieval performance that is competitive with dense retrievers on the unseen subsets of the BEIR and MAIR zero-shot retrieval benchmarks. They also perform ablation experiments demonstrating and speculating on the benefit of their three main methods.

**Strengths:**

1. The paper presents 3 advancements to GR semantic docid generation, pseudo-query generation for indexing, and retrieval decoding that are reasonably novel, convincingly better than previous GR methods, and have a high chance of informing future methods, even if their specific instantiations are not kept.

2. The paper’s narrative is well-motivated and clear to follow.

**Weaknesses:**

1. Several key empirical claims comparing to dense retrievers are overstated or unsupported by the presented results:
- L66-67: “Our best model … surpasses strong dense retrieval … including BEIR”
  - It is beaten on BEIR by even E5 and BGE base models which have >10x fewer params
  - A rough interpolation of the performance frontier defined by DRs in Fig 5 would put InstructGR on or below, not surpassing.
- L338-339 (note that this is in the MAIR eval section 6.1): “Despite having a relatively modest model size of 3B parameters, ZEROGR delivers competitive or even superior results compared to larger instruction-tuned dense retrievers with 7B parameters.”
  - Beaten on MAIR avg by >5 points by both 7B DRs, no MAIR domains where it is superior.

2. Evaluations (BEIR and MAIR) omit large-scale (>1M docs) retrieval settings, a known limitation of GR methods, but this scope restriction is not explicitly acknowledged, particularly in direct comparison with DRs.
- This and the previous issue could be addressed with some remarks comparing the favorable/unfavorable properties of GR vs DR.

3. Some figures and tables could be made clearer with minimal changes:
- Table 2 would benefit from bolding best results for readability, though care should be taken given the incomparability of model sizes
- Table 3 has no column or feature that allows us to distinguish prior methods (e.g. year, model size, docid/indexing/decoding methods). While the changing architectures might make the last option infeasible, anything would help.
- The grey lines vs bolded black lines in Figure 3 middle and right are not explained in the caption or discussion. I assume they are per-task accuracies from the “average” on L449, but this should be clearer.
- Figure 4’s inconsistent y axis scales implicitly equate differences of 3.7 acc@1, 2 nDCG@10, and 8.5 recall@100

4. Typos / Grammar / Formatting:
- L65: sentence ends in two periods.
- L351: “we can see our method achieve best performance among most datasets.” -> should be achieves
- Listing of docids methods in section 6.4 have the numerals separated from the method by a newline - unlucky alignment that could be addressed by a few extra words in the paragraph or removing the numerals entirely.

**Questions:**

1. Dependence between docid generation scheme and reverse-annealed decoding strategy:
- How does the docid generation scheme affect the trie structure (i.e. avg. # decodable tokens at each step), and does that indicate which docid generation strategies might suit which decoding strategies better?
- Is the docid generation strategy held constant (i.e. ZeroGR’s model) when ablating the decoding strategy in section 6.6 / figure 4, or are they based on prior work? E.g. Tay et al. 2022 uses hierarchical numerical docids with beam search decoding.

2. Task vs domain generalization:
- How do the domain distributions (categories and relative weight/quantity) differ between BEIR, MTEB, and MAIR?
- When scaling/choosing the training data, which would have the greatest marginal benefit:
  - An identical task (e.g. counterargument retrieval) in an unseen domain
  - A different task in an already-seen domain

 That is, how to disentangle the claim of quantity & diversity of *tasks* from the naturally better domain match of MAIR-Train (= ZeroGR-Train) which has all the domains in the eval set (by being from the same benchmark) vs BEIR and MTEB which may have sub/supersets of MAIR domains.

---

> ### Author Response · Authors · 2025-11-21
>
> Thank you for your insightful and constructive comments. We are encouraged to hear that you find our model design reasonable, well-motivated, and easy to follow. We address your concerns as follows:
>
> ---
>
> > **Several key empirical claims comparing to dense retrievers are overstated or unsupported by the presented results:**
>
> Thanks for pointing this out! We have revised our paper to correct these claims. Specifically, we have updated the related statements in Sections 1 and 6 to ensure they are more precisely scoped. Specifically, we clarify that our model *still underperforms large embedding models such as GritLM-7B and E5-Mistral-7B* on MAIR, and *outperforms several baselines such as BM25, Contriever, GTR, and GritLM-7B, but still underperforms other dense retrieval methods* on BEIR.
>
> ---
>
> > **Evaluations (BEIR and MAIR) omit large-scale (>1M docs) retrieval settings, a known limitation of GR methods, but this scope restriction is not explicitly acknowledged, particularly in direct comparison with DRs.**
>
> Thanks for your insightful comment. We acknowledge that our evaluation has focused on a modest-size corpus. Evaluating on a large-scale corpus would require substantially more computation and introduce model capacity limitations when handling large data. We have revised the paper to explicitly note this limitation in both the experimental setup and the conclusion.
>
> Notably, we would like to clarify that this paper focuses on enhancing **zero-shot retrieval ability** of generative retrieval on heterogeneous tasks, which is an understudied yet important challenge in GR. Handling large-scale corpora is a known limitation of GR with extensive prior work dedicated to it, but this falls outside the scope of our study. We also note that retrieval over modest-size heterogeneous corpora has broad practical relevance, such as in RAG systems, which is the scenario this work targets.
>
> ---
>
> > **Some figures and tables could be made clearer with minimal changes. Typos / Grammar / Formatting.
> Thanks for your comments! We have revised our paper to: (1) Improve readability of Table 2; (2) include detailed model information in Table 9; (3) Clarify the gray line in Figure 3; (4) Add note in Figure 4 to avoid mislead; (5) Fix the suggested grammar, typos, formatting issues.**
>
> > **How does the docid generation scheme affect the trie structure (i.e. avg. # decodable tokens at each step), and does that indicate which docid generation strategies might suit which decoding strategies better?**
>
> We thank the reviewer for this suggestion. We have added a figure **(Figure 9)** reporting the average number of decodable tokens at each step of the trie for both RQVAE and title based docids. The results show that title based docids create a trie with a very large branching factor at the first step (e.g., many more decodable tokens at step 1 than RQVAE), followed by a sharp reduction in later steps, while RQVAE produces a more gradually narrowing trie that becomes almost deterministic after 2–3 steps.
>
> This difference implies that title based docids can potentially benefit more from decoding strategies that encourage early diversity, whereas RQVAE is more naturally aligned with locally pruned decoding that exploits its quickly narrowing search space. At the same time, the average number of visited tokens per query is very similar for the two schemes (37.66 vs 39.25), indicating that these structural differences do not significantly affect overall decoding cost.
>
>
> ---
>
> > **Is the docid generation strategy held constant (i.e. ZeroGR’s model) when ablating the decoding strategy in section 6.6 / figure 4, or are they based on prior work? E.g. Tay et al. 2022 uses hierarchical numerical docids with beam search decoding.**
>
> Yes, the evaluated methods in Figure 4 employ the same docid generation strategy (i.e. generated by our docid generator) and same trained generative retrieval checkpoints, to ensure direct comparison.
>
> ---
>
> > **How do the domain distributions (categories and relative weight/quantity) differ between BEIR, MTEB, and MAIR?**
>
> Thanks for the question. We have added **Table 6** to illustrate the domain distribution of different datasets.
>
> ---
>
> (continue)

---

> ### Author Response · Authors · 2025-11-21
>
> ---
>
> > **When scaling/choosing the training data, which would have the greatest marginal benefit:
> An identical task (e.g. counterargument retrieval) in an unseen domain
> A different task in an already-seen domain
> That is, how to disentangle the claim of quantity & diversity of tasks from the naturally better domain match of MAIR-Train (= ZeroGR-Train) which has all the domains in the eval set (by being from the same benchmark) vs BEIR and MTEB which may have sub/supersets of MAIR domains.**
>
> An identical task in an unseen domain may be more beneficial. In addition, we believe performance is more directly related to the **query type** (e.g., short vs. long queries) and the **document type** (e.g., tables, code) rather than the domain itself, although domain can influence both. This is because in our work, the training data (query–doc pairs) is only used to train the **query generators** (note that the title generator is distilled from ChatGPT), and the final retrieval model is indexed using the generated query–docid pairs. Consequently, the retrieval model may face a smaller input distribution shift when the generated queries more closely resemble the inference-time queries. To ensure that query generators can produce such queries, it is helpful to include similar data in their training sets.
>
> In the following (also included as **Table 13** in the new PDF), we list the query types and document types in our training and evaluation data. We observe that 9 query types and 14 document types are shared across both sets. Beyond these, the test set contains 9 query types and 8 document types that do not appear in the training data.
>
> | **Category**        | **Query Types**                                                                                                                                       | **Doc Types**                                                                                                                                                     |
> | - | - | - |
> | **Train ∩ Eval**    | Question, Dialog, Claim, Function Header, NL Command, Code Problem, Math question, Paper Title, Summary *(9 types)*                           | Document, Answer, Function, Command Doc, Solution, Article, Articles, Medical Document, Paragraph, Pages, Statute, Passage, Passages, Table & Paragraph *(14 types)* |
> | **Only in Eval**    | Health Record, Topic, Situation, Request, Patient Data, Medical Case, Patient Description, Medical Claim, Numerical Claim *(9 types)*          | Clinical Trials, Prior Case, Communications, Dataset, Music, Tweet, News, POI, Table *(8 types)*                                                            |
> | **Only in Train**   | Math Statement, Entity & Relation, Paper Abstract, Entity Mention, CNL Command, GitHub Issue, Commit, Code Context, Math Question, Title, EU Directive, UK Legislation, Instruction, Reaction, Description *(14 types)* | Entity Page, Citation, Proof, Reference, Duplicate Question, Related File, Code Diff, Next Function, HuggingFace API, Tensor API, PyTorch API, UK Legislation, EU Directive, Highlight, Proteins Documents, Wikipedia Page *(16 types)* |
>
> ---
>
> **Thank you again for your thoughtful feedback! We hope these clarifications will resolve your concerns.**

---

> > ### Comment · Reviewer_Zbgo · 2025-11-25
> >
> > Thank you for the detailed response. These lead me to increase my score.

---

> ### Author Response · Authors · 2025-11-25
>
> Thank you for the confirmation and for considering an increased score! Your feedback is invaluable for improving our paper.

---

### Official Review · Reviewer_q9R8 · 2025-10-31

**Soundness:** 3
**Presentation:** 3
**Contribution:** 2
**Rating:** 4
**Confidence:** 3

**Summary:**

This paper proposes ZeroGR, a zero-shot generative retrieval framework that extends traditional generative retrieval (GR) to diverse information retrieval (IR) tasks using natural-language instructions. The framework includes three components: (1) a docid generator that converts heterogeneous documents into unified identifiers; (2) an instruction-tuned query generator that creates diverse pseudo-queries for corpus indexing; and (3) a reverse annealing decoding strategy balancing precision and recall. The model is evaluated on the BEIR and MAIR benchmarks, showing moderate improvements over dense retrieval baselines such as E5 and BGE, and competitive results compared to instruction-tuned retrievers.

**Strengths:**

- The paper is clearly written and provides a well-structured presentation of the proposed framework.

- It explores a new angle for generative retrieval, combining instruction tuning and docid generation under a zero-shot setting.

- The idea of leveraging task-level instructions for corpus indexing and retrieval is conceptually sound and connects GR with recent trends in instruction-based dense retrieval.

**Weaknesses:**

- The experimental evaluation is not comprehensive. Most results are based on a single Llama-3B model, and scalability is discussed mainly through small-scale ablations rather than large-scale experiments or efficiency analyses.
- The performance improvements are modest, often within 1–2 points over strong baselines, and sometimes lower on BEIR tasks. The claim of achieving state-of-the-art zero-shot performance is therefore not fully solid.
- Critically, the paper does not directly compare with existing generative retrieval baselines such as [1,2,3]. These methods are closest in formulation, and the lack of head-to-head evaluation makes it difficult to assess ZeroGR’s actual progress

[1] Learning to Tokenize for Generative Retrieval.

[2] Scalable and Effective Generative Information Retrieval. SIGIR 2024

[3] Exploring Training and Inference Scaling Laws in Generative Retrieval. SIGIR 2025

**Questions:**

- How does ZeroGR compare against other generative retrieval systems when evaluated under similar zero-shot conditions?
- How sensitive is the framework to the quality of instruction-tuned query generation?

Please refer to the weakness part.

---

> ### Author Response · Authors · 2025-11-21
>
> Thanks for your insightful comments! We are encouraged to hear that you find our work well-structured, exploring novel angles, and conceptually sound. We would like to address your concerns as follows:
>
> ---
>
> > **The experimental evaluation is not comprehensive. Most results are based on a single Llama-3B model, and scalability is discussed mainly through small-scale ablations rather than large-scale experiments or efficiency analyses.**
>
> Thanks for your comment. Indeed, our main model is based on Llama-3B. To evaluate a broader set of models, our ablation study in Figure 3 includes multiple model sizes (Qwen 0.5B, 1B, 3B) evaluated on five unseen tasks, where we observe consistent trends. While evaluating additional models would certainly provide further insights, running the full test suite (49 retrieval tasks) is computationally expensive, especially for larger LLMs that require multi-GPU parallel training. **We have updated the limitation section to reflect this and plan to evaluate larger models in future work.**
>
> ---
>
> > **The performance improvements are modest, often within 1–2 points over strong baselines, and sometimes lower on BEIR tasks. The claim of achieving state-of-the-art zero-shot performance is therefore not fully solid.**
>
>
> We acknowledge that our model does not achieve state-of-the-art performance on the overall benchmark. Our claim is that our method achieves the best results among generative retrieval approaches **(Table 3)** and outperforms strong dense retrieval baselines on multiple individual retrieval tasks **(Table 4)**. On average, we improve over the previous best generative retrieval method by about 3.8 points and achieve the top performance on most datasets. To further clarify our performance claims, we have revised **Section 6** to explicitly state that our gains are within the category of generative retrieval methods, along with a clearer comparison against dense retrieval.
>
> ---
>
> > **Critically, the paper does not directly compare with existing generative retrieval baselines such as [1,2,3]. These methods are closest in formulation, and the lack of head-to-head evaluation makes it difficult to assess ZeroGR’s actual progress.**
>
> > **How does ZeroGR compare against other generative retrieval systems when evaluated under similar zero-shot conditions?**
>
> Thanks for raising this point. We directly compare ZeroGR with [1] under the same zero-shot conditions in **Figure 3** and observe an average improvement of +3.8.
>
> In contrast, **[2] and [3] are designed for fully supervised settings** in which models are trained on static corpora (e.g., MS MARCO) with fixed DocID assignments. When evaluated on a new corpus like BEIR, these DocIDs no longer correspond to the new documents and preventing meaningful retrieval.
>
> Adapting these supervised methods to BEIR would require a BEIR-specific labeled training set and a new DocID assignment, which is exactly the bottleneck our method removes. In other words, techniques used in supervised generative retrieval (e.g., multi-stage training, distillation, ranking losses, curriculum strategies) are essentially orthogonal to our contribution. We intentionally use a simple SFT-based pipeline to keep ablations clean and isolate the effect of ZeroGR’s data generation and DocID construction. These additional training strategies could be incorporated in future work.
>
> ---
>
> > **How sensitive is the framework to the quality of instruction-tuned query generation?**
>
> The quality of instruction-tuned query generation plays a key role in training generative retrieval models. As shown in Figure 4, different query sources lead to noticeably different retrieval performance, and our generator produces more diverse queries that substantially improve accuracy.
>
> To isolate this effect, we conduct an ablation where we replace the doc2query-generated queries in the baseline TIGER (RQ-VAE) model with queries from our generator. As reported in the following table (also uploaded as **Table 11** in the paper), using our queries alone yields an overall improvement of **+6.4**. The gains are especially pronounced on domain-specific benchmarks, including legal (BillSum +30.0), finance (FinQA +26.7, TAT-DQA +20.0), code (LeetCode +24.0, CodeSearchNet +23.0), and scientific retrieval (Competition-Math +21.0, LitSearch +19.0). These results indicate that high-quality, diverse queries are crucial, particularly for specialized retrieval tasks.
>
> (continue)

---

> > ### Author Response · Authors · 2025-11-21
> >
> > | **Task**               | **doc2query** | **our query generator** | **Diff** |
> > |------------------------|---------------|---------------------------|----------|
> > | AILA2019-Case          | 2.00          | 2.00                      | +0.00    |
> > | Apple                  | 13.70         | 5.48                      | -8.22    |
> > | ArguAna                | 12.00         | 11.00                     | -1.00    |
> > | BillSum                | 36.00         | 66.00                     | **+30.00** |
> > | ClinicalTrials_2021    | 4.67          | 6.67                      | +2.00    |
> > | ClinicalTrials_2023    | 1.35          | 2.70                      | +1.35    |
> > | CodeEditSearch         | 13.00         | 22.00                     | +9.00    |
> > | CodeSearchNet          | 33.00         | 56.00                     | **+23.00** |
> > | Competition-Math       | 40.00         | 61.00                     | **+21.00** |
> > | Conala                 | 3.00          | 9.00                      | +6.00    |
> > | ConvFinQA              | 22.92         | 37.50                     | +14.58   |
> > | FiQA                   | 7.00          | 13.00                     | +6.00    |
> > | FinQA                  | 14.44         | 41.11                     | **+26.67** |
> > | LeetCode               | 6.00          | 30.00                     | **+24.00** |
> > | LegalQuAD              | 10.00         | 4.00                      | -6.00    |
> > | LitSearch              | 12.00         | 31.00                     | **+19.00** |
> > | NFCorpus               | 41.00         | 6.50                      | -34.50   |
> > | News21                 | 13.67         | 21.88                     | +8.20    |
> > | SciDocs                | 16.00         | 14.00                     | -2.00    |
> > | SciFact                | 34.00         | 42.00                     | +8.00    |
> > | StackMathQA            | 13.00         | 26.00                     | +13.00   |
> > | TAT-DQA                | 7.14          | 27.14                     | **+20.00** |
> > | ToT_2023               | 3.00          | 0.00                      | -3.00    |
> > | TopiOCQA               | 18.00         | 8.00                      | -10.00   |
> > | Touche                 | 46.94         | 39.80                     | -7.14    |
> > | **Average**            | **16.95**     | **23.35**                 | **+6.40** |
> >
> > ---
> >
> > **Thank you again for your thoughtful feedback! We hope these clarifications will resolve your concerns.**

---

### Official Review · Reviewer_cqwj · 2025-11-01

**Soundness:** 3
**Presentation:** 2
**Contribution:** 4
**Rating:** 6
**Confidence:** 4

**Summary:**

The paper presents ZeroGR, a novel framework for zero-shot generative retrieval that unifies document indexing, pseudo-query generation, and decoding under a single generative paradigm. It introduces three coordinated modules: (1) a DocID Generator that converts heterogeneous documents into short, interpretable textual identifiers; (2) an Instruction-Tuned Query Generator that produces diverse pseudo-queries to enrich training coverage; and (3) a Generative Retriever trained to map queries directly to DocIDs through maximum-likelihood learning. During inference, ZeroGR employs a reverse-annealing decoding strategy within a Trie-constrained search space, balancing precision and recall by gradually increasing the sampling temperature. The method is evaluated on MAIR and BEIR benchmarks, demonstrating robust zero-shot generalization and outperforming several state-of-the-art dense and generative baselines across seen and unseen retrieval tasks.

**Strengths:**

- Introduces a novel approach to extending generative retrieval into the zero-shot setting, demonstrating strong generalization through instruction-tuned pseudo-queries that transfer effectively across tasks.
- Presents a reverse-annealing decoding strategy that balances precision and diversity, leading to consistent improvements in both accuracy and recall.
- Conducts extensive evaluations on MAIR and BEIR, supported by thorough ablation studies that confirm the robustness and reliability of the proposed method.

**Weaknesses:**

1. Most of baselines originally trained on smaller corpora (e.g., NQ-320k or MS MARCO) rather than on the larger-scale MAIR dataset used for ZeroGR. Given that Section 6.3 itself demonstrates substantial performance gains from scaling training data and task diversity, the comparison may not be fully fair. Similar issues exist for dense retriever baselines (e.g., Contriever trained on MS MARCO). Reporting the precise training configurations and data sources for all baselines would clarify the relative contribution of ZeroGR.

2. The experimental setup for some analyses remains insufficiently described. For instance, Section 6.4 on DocID design does not clearly state which search or decoding configurations were used during evaluation—such as whether reverse-annealing, Trie constraints, and top-k generation parameters were fixed across variants. Likewise, the section 6.6 does not report the beam size used for beam search or the specific hyperparameters applied in the reverse-annealing schedule. These omissions make it difficult to assess the fairness, reproducibility, and stability of the reported comparisons.

Minor presentation issues: figures in appendix and abstracts in openreview still refer to the model as InstructGR, and a few figures appear to have untrimmed borders or inconsistent formatting.

**Questions:**

Could the authors clarify which datasets and training configurations were used for each generative and dense baseline in Tables 2–3? Would the authors consider retraining or fine-tuning the baselines on comparable data scales to ensure a fair comparison and isolate the methodological gains of ZeroGR?

Could the authors provide more details on decoding and search configurations in Sections 6.4 and 6.6—specifically the beam size, top-k/top-p settings, and reverse-annealing parameters, and confirm whether these were kept consistent across variants?

What is the exact composition of the MAIR dev / unseen-dev subsets used in ablations (e.g., task list and query counts)? This would help improve reproducibility.

Since the DocID length is fixed in the prompt, have the authors tested whether varying this length influences retrieval accuracy or DocID conflict rates?

---

> ### Author Response · Authors · 2025-11-21
>
> Thank you for your insightful and valuable comment! We are encouraged to hear that you found our approach novel, the results strong, and the evaluation extensive. Thank you for your insightful and valuable comment! We are glad to hear that you found our approach novel, the results strong, and the evaluation extensive.
>
> ---
>
> > **Most of baselines originally trained on smaller corpora rather than on the larger-scale MAIR dataset used for ZeroGR. Given that Section 6.3 itself demonstrates substantial performance gains from scaling training data and task diversity, the comparison may not be fully fair. Similar issues exist for dense retriever baselines. Reporting the precise training configurations and data sources for all baselines would clarify the relative contribution of ZeroGR.**
>
> > **Could the authors clarify which datasets and training configurations were used for each generative and dense baseline in Tables 2–3? Would the authors consider retraining or fine-tuning the baselines on comparable data scales to ensure a fair comparison and isolate the methodological gains of ZeroGR?**
>
> Thank you for the suggestion! We have added **Table 8** to illustrate the training configurations of the baseline methods. We also note that for ZeroGR, the training data is only used to train the title generator and the query generator, while the final generative retrieval model is initialized directly from the base LLM (i.e., Llama-3B).
>
> Your suggestion of retraining baselines on comparable data is valuable. We attempted to further fine-tune the E5-Large model on our ZeroGR-Train datasets with an in-batch InfoNCE loss but observed degraded performance. We therefore suspect that using ZeroGR-Train to train the dense retrieval model may require additional optimization, although it remains effective for training strong query and title generators in our work.
>
> ---
>
> > **The experimental setup for some analyses remains insufficiently described. For instance, Section 6.4 on DocID design does not clearly state which search or decoding configurations were used during evaluation—such as whether reverse-annealing, Trie constraints, and top-k generation parameters were fixed across variants. Likewise, the section 6.6 does not report the beam size used for beam search or the specific hyperparameters applied in the reverse-annealing schedule. These omissions make it difficult to assess the fairness, reproducibility, and stability of the reported comparisons.**
>
> > **Could the authors provide more details on decoding and search configurations in Sections 6.4 and 6.6—specifically the beam size, top-k/top-p settings, and reverse-annealing parameters, and confirm whether these were kept consistent across variants?**
>
> Thank you for this valuable suggestion! For analysis experiments, unless explicitly stated otherwise, we use a fixed decoding setup: reverse annealing decoding with trie constraints and fixed hyperparameters. For the experiment in Section 6.6, the beam size is 100, and the remaining hyperparameters match those in the main experiments. For all generations, we apply no top-k or top-p pruning, and the reverse annealing parameter is kept consistent across datasets and models (see Figure 10 for implementation details and default hyperparameters). We have updated the implementation details section to include these clarifications.
>
> ---
>
> > **Minor presentation issues: figures in appendix and abstracts in openreview still refer to the model as InstructGR, and a few figures appear to have untrimmed borders or inconsistent formatting.**
>
> Thank you for pointing this out! We have revised all problematic figures to ensure consistent terminology.
>
> ---
>
> > **What is the exact composition of the MAIR dev / unseen-dev subsets used in ablations (e.g., task list and query counts)? This would help improve reproducibility.**
>
> Thanks for your suggestion! We have added a new table (Table 10) that presents the composition of the dev subsets used in our ablations.
>
> ---
>
> > **Since the DocID length is fixed in the prompt, have the authors tested whether varying this length influences retrieval accuracy or DocID conflict rates?**
>
> Thanks for your question! In Figure 2 (middle), we have an analysis of document-level conflicts, and the observed conflict rate is very low (0.639%). To further clarify this, we have added an additional Figure 8 that evaluates conflict rates when treating only a prefix of the generated DocID as the identifier. The results show that the conflict rate drops below 1% once the prefix length exceeds 6 words, and with an 8-word prefix the conflict rate is only 0.45%. This confirms that our default DocID length is sufficient and that retrieval accuracy is not meaningfully impacted by DocID length.
>
> ---
>
> **Thank you again for your thoughtful feedback! We hope these clarifications will resolve your concerns.**

---

### Official Review · Reviewer_NVdj · 2025-11-02

**Soundness:** 3
**Presentation:** 3
**Contribution:** 2
**Rating:** 4
**Confidence:** 4

**Summary:**

The authors proposed a method for zero-shot generative retrieval. First given the document corpus and an instruction that encodes the task, document IDs for each documents are generated. Then a  query generator generates queries for each document for generative retrieval training. The authors showed competitive performance against other dense retrieval or generative retrieval methods for BEIR and MAIR benchmarks.

**Strengths:**

- Relatively simple method that the authors have shown to be working well
- Highlighted the importance of instruction-based retrieval

**Weaknesses:**

- There exists simpler methods for unified docID representation that are not compared against, see Questions below.
- Sec 6.2 on BEIR: Seems that more recent dense retrieval methods outperform generative retrieval methods here. Please explain.

**Questions:**

- According to the prompt in Appendix A, one could envision a much simpler way of deriving the docID: rank the terms in the document and select top-k by IDF value. Would this perform well?
- Please clarify how reverse-annealed generation help with beam search: since it is a temperature-based scaling method, it should not modify the relative ranking of the hypotheses in the beam search -- so the top-K from the beam search should be identical?

---

> ### Author Response · Authors · 2025-11-21
>
> Thank you for your valuable comment. We are glad to hear that you find our work simple yet effective. We would like to address your concerns as follows:
>
> ---
>
> > **There exists simpler methods for unified docID representation that are not compared against, see Questions below. According to the prompt in Appendix A, one could envision a much simpler way of deriving the docID: rank the terms in the document and select top-k by IDF value. Would this perform well?**
>
> Thank you for this suggestion. Using top-k IDF terms as a docID, while simple, has several drawbacks. First, the selected terms form an **unordered** bag of words, which breaks the sequential structure required by language models. Second, IDF-based selection **does not generalize to other modalities** such as tables or code, where “terms” are not well defined. Third, IDF depends on global corpus statistics, making the docID **non-deterministic** — adding or removing unrelated documents can change the IDF values and thus modify an existing document’s docID.
>
> Our title-based generator does not suffer from these issues. It produces a coherent, ordered, semantically meaningful string that aligns with the LM’s generative capabilities and remains stable across corpora. Below we show examples of different types of DocIDs (also included as **Table 12** in the revised PDF).
>
> | **Type**  | **Example** |
> | - |  - |
> | Random | asd8xc2c9ma90xj2398 |
> | Sentence  | LIMASSOL, Cyprus, April 28, 2021 /PRNewswire/ -- One of the top financial investment firms of the FX industry, Windsor Brokers |
> | Query | Induction of myelodysplasia by myeloid-derived suppressor cells. |
> | RQ-VAE |  g16289 g13509 g10485 g11274 g369 g3661 g13026 g8187 |
> | IDF      | brokerswindsor mt4 brokerswere kontos windsorbrokers |
> | Ours  | rna folding computational methods thermodynamic optimization model |
>
> ---
>
> > **Sec 6.2 on BEIR: Seems that more recent dense retrieval methods outperform generative retrieval methods here. Please explain.**
>
> We partially agree with this observation. We believe the strong performance of SOTA dense retrieval models on BEIR can be explained by two primary factors.
>
> * First, many BEIR tasks have large-scale training data that modern embedding models have been trained on, whereas MAIR is a newer diverse benchmark where these models generalize less effectively. That may explain why dense retrieval has more advantage in BEIR.
>
> * Second, dense retrieval has benefited from years of targeted optimization for zero-shot evaluation in BEIR  since 2021, including techniques like hard-negative mining, distillation, and pre-training [R1, R2, R3]. In contrast, our work is among the first to improve zero-shot generative retrieval performance across many heterogeneous tasks, with a simple training strategy (i.e., Eq. 4). Therefore, a performance gap from the SOTA dense systems is expected. Still, our approach significantly narrows this gap relative to existing generative retrieval methods (Table 3 and Figure 5).
>
> In addition, although our method includes a query generator trained on diverse data, the final retrieval model itself is not trained on any supervised IR datasets, it is initialized from Llama-3B-Instruct and trained only on synthetic data generated by the query and docid generators. We believe that additional pre-training on IR datasets, or applying more advanced techniques such as hard-negative mining and distillation, could further reduce the remaining gap.
>
> [R1] Approximate nearest neighbor negative contrastive learning for dense text retrieval
> [R2] In-Batch Negatives for Knowledge Distillation with Tightly-Coupled Teachers for Dense Retrieval
> [R3] Text embeddings by weakly-supervised contrastive pre-training
>
> ---
>
> > **Please clarify how reverse-annealed generation help with beam search: since it is a temperature-based scaling method, it should not modify the relative ranking of the hypotheses in the beam search -- so the top-K from the beam search should be identical?**
>
> Thanks for your question. We would like to clarify that our method is **not a supplement** to beam search but **a different stochastic sampling algorithm**, as detailed in **Algorithm 1**.
>
> Beam search greedily preserves the top-K highest-probability continuations at each step, which severely limits diversity: candidates converge to the same prefix and differ only in minor suffixes. In contrast, our reverse-annealed decoding gradually increases the sampling temperature and expands candidates using this softened distribution rather than deterministically selecting top tokens. This allows the search to generate lower-probability hypotheses that greedy beam search would consistently prune, enabling it to escape local optima and explore a broader solution space. As shown in Figure 4, this produces a much more diverse set of docID candidates and leads to a clear improvement in recall.
>
> ---
>
> **Thank you again for your thoughtful feedback! We hope these clarifications will resolve your concerns.**

---

### Author Response · Authors · 2025-11-21

**Thank all reviewers for their insightful and constructive comments!**

We are encouraged to learn that you recognize the clarity and simplicity of our framework, the conceptual soundness of our formulation, and the novelty of our contributions in query / docid design, and decoding algorithms. We also appreciate your positive comments on the quality of our presentation and the thoroughness of our evaluations and ablations, which confirm the robustness and generalization ability of our method.


We appreciate the invaluable suggestions that help us improve our paper. In light of these comments, we have addressed the following main concerns with corresponding revisions:
* **Performance comparison to dense retrieval (Reviewers NVdj, q9R8, Zbgo).** We have revised **Sections 1, 6, and 7** (changes marked in red) to more accurately position our comparison with dense retrieval and updated **Tables 2 and 3** for better readability. We would like to clarify that dense and generative retrieval differ fundamentally in model architecture, training data, and optimization setups; our goal is to improve the zero-shot capability of generative retrieval through simple, controlled designs rather than to pursue absolute benchmark SOTA against dense retrieval.
* **Missing detailed training and inference configurations for dense and generative baselines (Reviewers Zbgo, cqwj).** We have added **Table 6** containing dataset statistics, **Table 8** detailing dense retrieval configurations, and **Table 9** summarizing generative retrieval configurations.
* **Evaluation on larger corpora and larger models (Reviewers q9R8, Zbgo).** We have revised **Sections 6 and 7** to acknowledge this limitation. We would like to clarify that our focus is on improving zero-shot generalizability of generative retrieval, primarily through better query and docid generators. Scaling to larger corpora is orthogonal and a promising direction for future work.
* We have also added Figures 8 and 9 for additional docid analysis (Reviewers Zbgo, cqwj) and corrected typos and formatting issues (Reviewer Zbgo).

**We have revised the paper accordingly and uploaded a new PDF with all changes marked in red.**

---

### Meta-Review · Area_Chair_mEjh · 2026-01-07

**Summary:**

This paper proposes a framework for zero-shot generative retrieval which does not require training data from new domains. They do this by training two small models (1B) to generate docids and synthetic queries from each of the documents in a new corpus. These two components are trained on the MAIR benchmark’s training subsets. The outputs of these components are then used to train another small model (ZeroGR) to output a ranked set of docids based on a query using reverse-annealing and a trie structure based on the indexed corpus. They show that this framework outperforms other GR methods in similar parameter scales.

Most of the concerns were addressed by the authors in the rebuttal. The main outstanding concerns are the following:

- Table 12 should contain several examples of documents and the set of different types of docids in order to appropriately evaluate their meaningfulness.
- The authors should add the scale of each training dataset in Tables 8 and 9 to make the training settings easier to understand and compare.
- The rest of the concerns such as toning down their claims and clarifying their methodology were appropriately addressed.

**Reviewer Concerns:**

R1:
- Try baseline using terms in document ranked by IDF value
	- The authors ignore this suggestion, saying that this docid type will not be meaningful and providing an example in Table 12 to show this. Although I agree with the authors to some extent, Table 12 should contain different type of docids for the same document to avoid cherry-picking meaningful or non-meaningful decides from different categories.
- Explain why recent dense retrieval methods outperform generative retrieval on BEIR
	- Dense retrieval methods have large training datasets and years of compounding techniques applied to zero-shot evaluation in BEIR so a difference in performance is expected compared to ZeroGR’s new model class.
- Why does reverse-annealed generation help with beam search?
	- The reverse-annealed generation replaces beam search and just allows for a more diverse set of docids to be created.

R2:
- Differing training datasets make fair method comparisons difficult
	- The authors added Table 8 to specify all of the known training datasets for each dense retrieval method. I believe this reviewer would have appreciated the inclusion of the size of each dataset in this table but this was not added.
	- Additionally, the authors claim that further fine-tuning E5-Large on their training data led to degraded performance.
	- The difficulty in fair comparison remains with dense retrieval methods but the comparison with GR methods seems valid.
- Methodological and hyperparameter clarifications are necessary for acceptance
	- Their experiments only use one setup: reverse annealing decoding with trie constraints and fixed hyperparameters. All of these details were added to the paper.
- What are the statistics of the MAIR dev/unseen-dev subsets?
	- Table 10 added to paper.
- Does varying the length of the DocID influence performance?
	- The authors added an experiment which replaces each docid with a prefix of varying lengths and there are very small changes in the already very low conflict rates (% of documents assigned the same docid) above 6 tokens. This seems to show the method’s robustness to the current docid length constraints.

R3:
- Experimental evaluations are not comprehensive enough (only one model and scalability is not well-tested)
	- Resource constraints bar the author’s capacity to run Qwen models on the full test suit but the small scale Qwen experiments show similar trends. This is a minor concern.
- Performance improvements are modest and so SOTA claims are not well-substantiated.
	- Authors concede and will tone-down claims.
- No head-to-head comparison with three recent generative retrieval baselines.
	- One of the baselines was compared with ZeroGR in the original paper and the other two are fully supervised baselines that are not easily transferred between one setting to another (exactly due to the challenge that ZeroGR addresses).
- How sensitive is the method to the quality of instruction-tuned query generation?
	- The authors conduct an ablation which shows that their queries improve the performance of a different method considerably, showing that the quality and diversity of their generated queries is crucial.

R4: (The reviewer claims that their concerns were addressed and they wanted to raise their score)
- Claims of improved performance are overstated
	- Authors concede and will tone-down claims.
- Scope restriction to <1M docs is not acknowledged
	- Authors acknowledge this limitations but claims that this is a known GR issue and their method focuses on tackling zero-shot retrieval ability which is an understudied challenge for GR methods.
- Table improvements and small corrections
	- All the style improvements suggested were included in the paper.
- Dependence between docid generation and reverse-annealed decoding strategy
	- Added an analysis of the relationship between docid generation methods and the number of decodable tokens at each step.
- Task & domain generalization
	- The authors added a table listing the query and document types that are unique to their training and evaluation subsets.

**Reviewer Scores:**

- NVdj 4 -> 4
	- I think the reviewer was unlikely to change their score since they believed the contribution to be only fair and the rebuttal did not address this.
- cqwj 6 -> 6
	- The rebuttal was likely to leave positive score.
- q9R8 4 -> 6
	- The rebuttal was strong and the reviewer does not seem very confident in their initial score. Some of the reviewers concerns were somewhat weak to begin with.
- Zbgo 4 -> 6
	- Reviewer said they would raise their score.

---

### Decision · Program_Chairs · 2026-01-26

Accept (Poster)